# IDENTIFYING UNPERTURBED CELLULAR PROGRAMS ENABLES ACCURATE SINGLE-CELL PERTURBATION PREDICTION

## ABSTRACT

Predicting cellular responses to single/combinatorial gene perturbations is a central challenge in functional genomics. A critical limitation of current models is their inability, both theoretically and methodologically, to disentangle perturbation-induced effects from the pervasive background cellular transcriptional programs that remain invariant to perturbations but dominate observed gene expression patterns. To address this, we propose a latent variable generative model that explicitly partitions latent space into an variant subspace where a latent causal model is employed to capture perturbations, and an invariant subspace capturing unperturbed cellular programs. We establish a principled foundation for disentangling these two subspaces, and identifying the latent causal model, by differentiability analysis. We then translate our theoretical findings into a practical method that more accurately predicts perturbation effects, supported by the theoretical guarantees. On both simulated and large-scale genetic perturbation benchmarks, the proposed method achieves state-of-the-art accuracy in predicting cellular responses to unseen combinations, significantly outperforming existing methods. Crucially, by disentangling unperturbed cellular programs from perturbation-induced effects, our method prevents the latter from being confounded or absorbed into the dominant invariant patterns. This separation allows the true causal impact of perturbations to be isolated and reliably estimated, thereby enabling accurate prediction of unseen combinatorial gene perturbations at the single-cell level.

## 1 INTRODUCTION

Understanding the generative process that links genotype to cellular phenotype is a central challenge in modern biology and medicine (Orgogozo et al., 2015). A key experimental strategy toward this goal is systematic gene perturbation, where genes are perturbed and the resulting cellular phenotypes are measured. The emergence of CRISPR-based perturbation technologies has made such large-scale experiments feasible (Jinek et al., 2012; Gilbert et al., 2014; Dixit et al., 2016; Replogle et al., 2020). However, despite their transformative power, these approaches remain prohibitively expensive, time-consuming, and sometimes ethically constrained, making exhaustive screening across genes and perturbation combinations infeasible (Uddin et al., 2020; Caplan et al., 2015).

To overcome these experimental bottlenecks, recent studies have turned to machine learning, training models on observational and limited perturbation data to predict cellular outcomes under novel perturbations (Lin & Wong, 2018; Castillo-Hair et al., 2024; Lotfollahi et al., 2023; Rood et al., 2024; Szałata et al., 2024). Such models aim to generalize beyond available experiments, including to complex multi-gene perturbations that have never been observed. However, this is inherently difficult: it corresponds to prediction under distribution shift, where the test distribution (unseen perturbations) differs from the training distribution (observed perturbations). The challenge is magnified in the combinatorial setting, as multi-gene perturbations can induce far more severe distribution shifts than single-gene ones (Roohani et al., 2024).

**Related works.** One promising research direction to addressing distribution shift is to infer the causal mechanisms underlying the data, as models are inherently capable of predicting outcomes un-

der distribution shifts induced by interventions (e.g., gene perturbations)[1] (Pearl, 2009; Schölkopf, 2022). Adopting this perspective, recent work has formulated single-cell perturbation prediction using latent causal generative models (Lachapelle et al., 2022; Zhang et al., 2023; Lopez et al., 2022; de la Fuente et al., 2025), aiming to learn causal representations from observational and limited perturbation data. These learned representations correspond to the underlying latent causal mechanisms, an approach commonly referred to as causal representation learning (Schölkopf et al., 2021). Though conceptually promising, a fundamental question concerns identifiability guarantees: can the true latent causal mechanisms be uniquely recovered from observational and limited interventional data, up to a simple transformation? Very recently, theoretical results have begun to address this question (Lachapelle et al., 2022; Zhang et al., 2023), and building on this foundation, several methods have subsequently been proposed (Lopez et al., 2022; Zhang et al., 2023; de la Fuente et al., 2025). Additional related works, including disentangling perturbation effects, identifiable causal representations, and contrastive representation learning, are provided in App. A.

**Motivations.** However, current identifiability results generally assume access to such precious interventional data, in which all latent causal variables must have been perturbed (Liu et al., 2022; Varici et al., 2025; Liu et al., 2024)[2]. Such interventional data are rarely obtainable in real cellular experiments, as comprehensive perturbation of all genes is often prohibitively expensive; typically, only a small subset of genes can be experimentally perturbed (Replogle et al., 2022; Reymond, 2015). Consequently, a vast subspace of genes remains unperturbed. As a result, existing identifiability theory, which typically assumes access to interventional data for all latent causal variables, may not be directly applicable to real cellular datasets, and, in turn, methods built upon these theoretical results (Lopez et al., 2022; Zhang et al., 2023; de la Fuente et al., 2025) may also struggle to perform effectively in practice, given the limited and partial interventional data typically available.

**Contributions.** To address this critical gap, this paper makes the following contributions. *A New Generative Model (§ 2).* We introduce a novel latent variable model that explicitly partitions the latent space into two components: a causal subspace, capturing the perturbable portion of the gene space, and an invariant subspace, representing the unperturbed portion. *Identifiability Guarantees (§ 3).* We derive sufficient conditions for the identifiability of the causal model within the causal subspace, providing a key theoretical contribution that extends prior results (Lachapelle et al., 2022; Zhang et al., 2023). *A Practical Learning Framework (§ 4).* We translate our theoretical insights into a practical method, a general framework for learning both the latent causal variables in the causal subspace and their causal structure from single-cell data. *Extensive Empirical Validation (§ 5).* We conduct comprehensive experiments on single- and multi-gene perturbation benchmarks, showing that the proposed method significantly outperforms existing methods in predicting responses to unseen combinations and recovers biologically meaningful latent factors.

## 2 PROBLEM SETUP: A NOVEL LATENT CAUSAL GENERATIVE MODEL

In single-cell perturbation prediction, interventional data are typically available only for a small subset of genes. These data are generated through targeted gene perturbations followed by single-cell transcriptomic profiling, as exemplified by Perturb-seq (Dixit et al., 2016) and its direct-capture variants (Replogle et al., 2020). Exhaustively perturbing all genes is prohibitively expensive, necessitating modeling approaches that can effectively leverage limited-perturbation data. In this section, we formulate the problem using a latent causal generative modeling framework. Refer to App. B.1 for a summary of notation and a complete list of symbols used throughout the paper.

### 2.1 LATENT CAUSAL GENERATIVE MODELING UNDER LIMITED INTERVENTIONS

We now introduce a latent causal generative model, in which each cell is associated with an observed expression profile $\mathbf{x}$. These observed profiles are generated from an underlying latent space $\mathbf{z}$, which provides a compact representation of the cell's internal state. In particular, $\mathbf{z}$ captures both

---

[1] In the scope of this work, perturbations can be viewed as *interventions* in the causal sense, we thus use "perturbations" and "interventions" interchangeably throughout this paper.

[2] If some latent causal variables remain unperturbed, additional assumptions such as sparse graph structures (Lachapelle et al., 2022) are generally required, though often hard to justify in real cellular processes.

background cellular transcriptional programs—stable regulatory and transcriptional patterns largely unperturbed under experimental conditions—and perturbation-induced effects.

To model limited-perturbation scenarios, we split the latent space into two subspaces, as in Figure 1a:

- $\mathbf{z}_\iota$ (*perturbation-invariant block*), supported on $\mathcal{Z}_\iota \subseteq \mathbb{R}^{d_\iota}$, represents the invariant subspace corresponding to background programs, which are typically difficult or costly to perturb. Examples include donor genotype, stable chromatin context, and core regulatory programs.

- $\mathbf{z}_\nu$ (*perturbation-responsive block*), supported on $\mathcal{Z}_\nu \subseteq \mathbb{R}^{d_\nu}$, represents the variant subspace that is susceptible to perturbations, including features such as pathway activity, dose-response effects, and compensatory programs. The variant latent subspace $\mathbf{z}_\nu$ involves an unknown causal structure, constrained to follow a directed acyclic graph (DAG).

To formalize perturbations on $\mathbf{z}_\nu$, we introduce a surrogate variable $\mathbf{u} \in \mathcal{U}$ that identifies which perturbation has been applied (e.g., a one-hot encoding). We do not require knowledge of the specific intervention mechanism; it is sufficient to know that a perturbation has occurred. Each latent block is associated with independent exogenous variables: $\mathbf{n}_\iota$ for $\mathbf{z}_\iota$ and $\mathbf{n}_{\nu,i}$ for each coordinate of $\mathbf{z}_\nu$, capturing external sources of variation. Finally, all latent endogenous variables, $\mathbf{z}_\iota$ and $\mathbf{z}_\nu$, are combined through an unknown generative process to produce the observed expression profile $\mathbf{x}$.

Without further assumptions, the latent variables $\mathbf{z}_\iota$ and $\mathbf{z}_\nu$, and in particular the causal structure among $\mathbf{z}_\nu$, cannot, in general, be identified solely from the observed variables $\mathbf{x}$ and $\mathbf{u}$. To enable the theoretical analysis that follows, we parameterize the proposed causal generative model as follows.

$$\mathbf{z}_\iota := \boldsymbol{\lambda}_{\iota\iota}\,\mathbf{z}_\iota + \mathbf{n}_\iota, \qquad \mathbf{n}_\iota \sim \mathcal{N}\big(\boldsymbol{\mu}_\iota, \operatorname{diag} \boldsymbol{\beta}_\iota\big), \tag{1}$$

$$\mathbf{z}_\nu := \boldsymbol{\lambda}_{\nu\iota}(\mathbf{u})\,\mathbf{z}_\iota + \boldsymbol{\lambda}_{\nu\nu}(\mathbf{u})\,\mathbf{z}_\nu + \mathbf{n}_\nu, \qquad \mathbf{n}_\nu \sim \mathcal{N}\big(\boldsymbol{\mu}_\nu(\mathbf{u}), \operatorname{diag} \boldsymbol{\beta}_\nu(\mathbf{u})\big), \tag{2}$$

$$\mathbf{x} := g(\mathbf{z}), \tag{3}$$

where,

- $\mathbf{n}_\iota \in \mathbb{R}^{d_\iota}$ and $\mathbf{n}_\nu \in \mathbb{R}^{d_\nu}$ are latent exogenous variables, sampled from $\mathcal{N}\big(\boldsymbol{\mu}_\iota, \operatorname{diag} \boldsymbol{\beta}_\iota\big)$ with mean $\boldsymbol{\mu}_\iota$ and variance $\operatorname{diag} \boldsymbol{\beta}_\iota$, $\mathcal{N}\big(\boldsymbol{\mu}_\nu(\mathbf{u}), \operatorname{diag} \boldsymbol{\beta}_\nu(\mathbf{u})\big)$ with mean $\boldsymbol{\mu}_\nu(\mathbf{u})$ and variance $\operatorname{diag} \boldsymbol{\beta}_\nu(\mathbf{u})$, respectively.

- The intra-block square matrices, i.e., $\boldsymbol{\lambda}_{\iota\iota}$ and $\boldsymbol{\lambda}_{\nu\nu}(\mathbf{u})$, are strictly upper triangular, while the cross-block $\boldsymbol{\lambda}_{\nu\iota}(\boldsymbol{u})$, by construction, is consistent with a fixed acyclic order $\mathbf{z}_\iota \prec \mathbf{z}_\nu$.[3]

- In Eq. (3), $\mathbf{z} = (\mathbf{z}_\iota, \mathbf{z}_\nu)$ and $g$ denotes an unknown nonlinear mapping from $\mathbf{z}$ to $\mathbf{x}$.

### 2.2 THEORETICAL TARGET: IDENTIFIABILITY

Our aim is to establish *identifiability* for the proposed latent causal generative model, i.e., to determine under which conditions the latent variables and the causal structure among them can be uniquely recovered from observational variables $\mathbf{x}$ and $\mathbf{u}$, up to a trivial transformation. Formally, we introduce the definitions as follows.

**Definition 2.1** (Block identifiability). Let $\mathcal{S} \subseteq \{1, \ldots, d_\iota + d_\nu\}$ index a subset of latent coordinates and $\mathbf{z}_\mathcal{S} \in \mathcal{Z}_\mathcal{S}$ its subvector. The block $\mathbf{z}_\mathcal{S}$ is *block-identifiable* via a representation map $f : \mathcal{X} \to \mathbb{R}^{|\mathcal{S}|}$ if the learned code $\hat{\mathbf{z}}_\mathcal{S} = f(\mathbf{x})$ is an invertible reparameterization of $\mathbf{z}_\mathcal{S}$ depending on no other latents. Formally, there exists a bijection $h : \mathcal{Z}_\mathcal{S} \to \mathbb{R}^{|\mathcal{S}|}$ with $\hat{\mathbf{z}}_\mathcal{S} = h(\mathbf{z}_\mathcal{S})$ a.s.

**Definition 2.2** (Component-wise identifiability). In the sense of Defn. 2.1, $\mathbf{z}_\mathcal{S}$ is *component-wise identifiable* if $h$ reduces to a per-coordinate affine transformation and permutation, i.e., there exist a permutation $\boldsymbol{P} \in \mathbb{R}^{|\mathcal{S}| \times |\mathcal{S}|}$, diagonal $\boldsymbol{D} \succ 0$, and vector $\mathbf{c} \in \mathbb{R}^{|\mathcal{S}|}$ such that $\hat{\mathbf{z}}_\mathcal{S} = \boldsymbol{P}\boldsymbol{D}\mathbf{z}_\mathcal{S} + \mathbf{c}$ a.s.

## 3 THEORY: IDENTIFIABILITY OF THE PROPOSED LATENT CAUSAL MODEL

We now state sufficient conditions under which the latent factors in § 2 are identifiable. Our analysis proceeds by (i) specifying mild structural and regularity assumptions on the latent SCM and the generative map $g$, (ii) defining a contrastive positive-pairing protocol aligned with limited interventions,

---

[3]Without loss of generality, we fix such an acyclic order across environments following Liu et al. (2022).

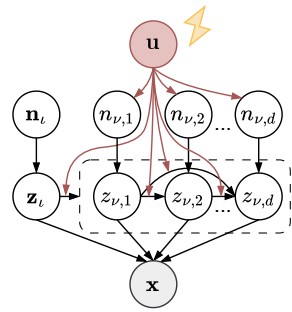

(a) Generative model of single-cell perturbations.

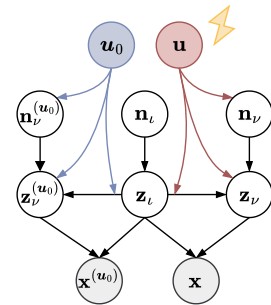

(b) Contrastive positive-pairing protocol.

Figure 1: Latent generative modeling. (a) Under perturbation identity $\mathbf{u}$, the perturbation-responsive factors $\mathbf{z}_\nu$ are influenced by $\mathbf{u}$ through latent mechanisms and their associated exogenous noises $\mathbf{n}$, while the invariant block $\mathbf{z}_\iota$ maintains unchanged. Together, the latent variables $\mathbf{z} := (\mathbf{z}_\nu, \mathbf{z}_\iota)$ generate the observed $\mathbf{x}$. (b) The invariant variables $\mathbf{z}_\iota$ are shared between the perturbed state $\mathbf{x}$ and its controlled counterpart $\mathbf{x}^{(\boldsymbol{u}_0)}$, while the responsive components differ as $\mathbf{z}_\nu$ and $\mathbf{z}_\nu^{(\boldsymbol{u}_0)}$, where $\boldsymbol{u}_0$ is a control setting for contrastive objective.

and (iii) proving that global maximizers of a joint likelihood-regularization objective recover $\mathbf{z}_\iota$ up to block reparameterization and $\mathbf{z}_\nu$ up to per-coordinate indeterminacies.

## 3.1 STRUCTURAL ASSUMPTIONS ON THE GENERATIVE MODEL

Under the generative model in Equations (1) to (3), we state technical assumptions for tractable theoretical analysis:

**Assumption 3.1** (Anchored weight-variant). *At the control $\boldsymbol{u}_0$, we have $\boldsymbol{\lambda}_{\nu\iota}(\boldsymbol{u}_0) = \mathbf{0}$ and $\boldsymbol{\lambda}_{\nu\nu}(\boldsymbol{u}_0) = \mathbf{0}$, which we regard as the baseline anchor.*

**Assumption 3.2** (Diffeomorphic generative mapping). *The generative map $g : \mathcal{Z} \to \mathcal{X}$ in Eq. (3) is a diffeomorphism, i.e., a $C^1$ bijection with a $C^1$ inverse.*

**Assumption 3.3** (Perturbation richness). *Fix a reference environment $\boldsymbol{u}_0 \in \mathcal{U}$. For each $j \in [d_\nu]$, let $\boldsymbol{\lambda}_j(\boldsymbol{u}) \in \mathbb{R}^{|\mathrm{pa}(j)|}$ denote the vector of incoming coefficients of $\mathrm{z}_{\nu,j}$ from its parents $\mathrm{pa}(j) \subseteq \{\mathbf{z}_\iota, \mathbf{z}_\nu\}$ that precede $j$ in the acyclic order. Write $\tau_j(\boldsymbol{u}) := \beta_{\nu,j}^{-1}(\boldsymbol{u})$ and $\kappa_j(\boldsymbol{u}) := \tau_j(\boldsymbol{u})\mu_{\nu,j}(\boldsymbol{u})$ for the Gaussian precision and natural mean of the noise of $\mathrm{z}_{\nu,j}$ under environment $\boldsymbol{u}$. We assume:*

*(a) There exists $\boldsymbol{u}_j$ such that the set $\{\boldsymbol{\lambda}_j(\boldsymbol{u}_j) - \boldsymbol{\lambda}_j(\boldsymbol{u}_0) : \boldsymbol{u}_j \in \mathcal{U} \setminus \{\boldsymbol{u}_0\}\}$ spans $\mathbb{R}^{|\mathrm{pa}(j)|}$.*

*(b) There exist $\boldsymbol{u}_j', \boldsymbol{u}_j'' \in \mathcal{U}$ such that the difference vectors $(\kappa_j(\boldsymbol{u}_j') - \kappa_j(\boldsymbol{u}_0), \tau_j(\boldsymbol{u}_j') - \tau_j(\boldsymbol{u}_0))$ and $(\kappa_j(\boldsymbol{u}_j'') - \kappa_j(\boldsymbol{u}_0), \tau_j(\boldsymbol{u}_j'') - \tau_j(\boldsymbol{u}_0))$ are linearly independent in $\mathbb{R}^2$.*

## 3.2 CONTRASTIVE POSITIVE-PAIRING PROTOCOL

We formalize how a positive pair is generated under the DGP in § 2 (see Figure 1b). Fix an anchor setting $\boldsymbol{u}_0 \in \mathcal{U}$ as in Asm. 3.1.[4] For each anchor cell, we pair a sample drawn under a randomly selected second perturbation setting $\mathbf{u} \sim q_\mathbf{u}$ on $\mathcal{U} \setminus \{\boldsymbol{u}_0\}$. When referring to the anchored perturbation setting $\boldsymbol{u}_0$, we denote the corresponding variables as $\mathbf{z}_\nu^{(\boldsymbol{u}_0)}$, $\mathbf{z}_\iota^{(\boldsymbol{u}_0)}$, $\mathbf{z}^{(\boldsymbol{u}_0)}$, and $\mathbf{x}^{(\boldsymbol{u}_0)}$ to emphasize their evaluation under $\boldsymbol{u}_0$. Otherwise, when variables are considered under a randomly selected perturbation setting, we use the general notations introduced in § 2.

**Assumption 3.4** (Perturbation excitation coverage). *For each coordinate $j \in [d_\nu]$, define the excitation set*

$$\mathcal{U}_j := \{\mathbf{u} \in \mathcal{U} \setminus \{\boldsymbol{u}_0\} : \boldsymbol{\lambda}_j(\mathbf{u}) \neq \boldsymbol{\lambda}_j(\boldsymbol{u}_0) \ \lor \ (\kappa_j(\mathbf{u}), \tau_j(\mathbf{u})) \neq (\kappa_j(\boldsymbol{u}_0), \tau_j(\boldsymbol{u}_0))\},$$

*where $\boldsymbol{\lambda}_j(\cdot), \tau_j(\cdot), \kappa_j(\cdot)$ are as in Asm. 3.3. Assume the second perturbation setting for each positive pair is drawn i.i.d. as $\mathbf{u} \sim q_\mathbf{u}$ on $\mathcal{U} \setminus \{\boldsymbol{u}_0\}$ with $q_\mathbf{u}(\mathcal{U}_j) > 0$ for all $j \in [d_\nu]$.*

---

[4]Any perturbation identity could serve as the anchor; w.l.o.g., we select $\boldsymbol{u}_0$ for notational clarity.

**Assumption 3.5** (Positive pairing protocol). *Fix an anchor $\boldsymbol{u}_0 \in \mathcal{U}$, and randomly sample $\mathbf{u} \sim q_{\mathbf{u}}$ on $\mathcal{U} \setminus \{\boldsymbol{u}_0\}$. For a sample $\mathbf{x}^{(\boldsymbol{u}_0)} = g(\mathbf{z}_\iota^{(\boldsymbol{u}_0)}, \mathbf{z}_\nu^{(\boldsymbol{u}_0)})$ under the control state $\boldsymbol{u}_0$, define the corresponding positive counterpart $\mathbf{x} = g(\mathbf{z}_\iota, \mathbf{z}_\nu)$ under perturbation $\mathbf{u}$. Assume the latent variables follow*

$$\mathbf{z}_\iota^{(\boldsymbol{u}_0)} = \mathbf{z}_\iota \sim p_{\phi^\circ}(\mathbf{z}_\iota), \qquad \mathbf{z}_\nu^{(\boldsymbol{u}_0)} \sim p_{\phi^\circ}(\mathbf{z}_\nu \mid \mathbf{z}_\iota^{(\boldsymbol{u}_0)}, \boldsymbol{u}_0), \qquad \mathbf{z}_\nu \sim p_{\phi^\circ}(\mathbf{z}_\nu \mid \mathbf{z}_\iota, \mathbf{u}),$$

*where, $p_{\phi^\circ}$ denotes the distribution generated by the latent SCM, with $\phi^\circ$ specifying the complete parameterization of the true data-generating process.*

### 3.3 IDENTIFIABILITY RESULTS

**Theorem 3.1** (Identifiability of the proposed latent causal generative model). *Consider smooth inference encoders $f : \mathcal{X} \to \mathbb{R}^{d_\iota + d_\nu}$, decomposed as $f(\mathbf{x}) = (f_\iota(\mathbf{x}), f_\nu(\mathbf{x}))$ with $\dim(f_\iota) = d_\iota$ and $\dim(f_\nu) = d_\nu$. Suppose Asms. 3.1 to 3.3 hold. Define the joint objective*

$$\mathcal{J}_{obj}(\phi, f) := \underbrace{\mathbb{E}_{(\mathbf{x}, \mathbf{u}) \sim p_{\phi^\circ}} [\log p_\phi(\mathbf{x} \mid \mathbf{u})]}_{Likelihood} - \alpha \underbrace{\mathbb{E}_{(\mathbf{x}^{(\boldsymbol{u}_0)}, \mathbf{x})} \left[ \|f_\iota(\mathbf{x}^{(\boldsymbol{u}_0)}) - f_\iota(\mathbf{x})\|_2^2 \right]}_{Alignment\ across\ \mathbf{u}}, \tag{4}$$

*where $\alpha > 0$ is a scaling constant, $(\mathbf{x}^{(\boldsymbol{u}_0)}, \mathbf{x})$ are positive pairs following Asm. 3.5, and $\mathbf{u} \sim q_{\mathbf{u}}$ as in Asm. 3.4. Let $(\phi^\star, f^\star)$ be a global maximizer of Eq. (4). At the global maximizer, the optimization is constrained so that for any $\mathbf{z}_\nu \in \mathcal{Z}_\nu$, the map $\mathbf{z}_\iota \mapsto f_\iota^\star \circ g(\mathbf{z})$ is injective.*

*Then, for any two global maximizers $(\phi^\star, f^\star)$ and $(\tilde{\phi}^\star, \tilde{f}^\star)$ that realize the true marginal $p_{\phi^\circ}(\mathbf{x}|\mathbf{u})$, i.e., $\mathbb{E}[\log p_{\tilde{\phi}^\star}] = \mathbb{E}[\log p_{\phi^\star}] = \mathbb{E}[\log p_{\phi^\circ}]$, the corresponding encodings satisfy:*

1. *(Block-identifiability of $\mathbf{z}_\iota$). There exist bijections $h_\iota, \tilde{h}_\iota : \mathcal{Z}_\iota \to \mathbb{R}^{d_\iota}$ such that $f_\iota^\star(\mathbf{x}) = h_\iota(\mathbf{z}_\iota)$ and $\tilde{f}_\iota^\star(\mathbf{x}) = \tilde{h}_\iota(\mathbf{z}_\iota)$ a.s., thus $\mathbf{z}_\iota$ is block-identifiable through $f^\star$ in the sense of Defn. 2.1.*

2. *(Component-wise identifiability of $\mathbf{z}_\nu$). There exist permutation $\boldsymbol{P} \in \mathbb{R}^{d_\nu \times d_\nu}$, diagonal $\boldsymbol{D} \succ 0$, and $\boldsymbol{c} \in \mathbb{R}^{d_\nu}$ such that $f_\nu^\star(\mathbf{x}) = \boldsymbol{P}\boldsymbol{D}\mathbf{z}_\nu + \boldsymbol{c}$ a.s.; likewise for $\tilde{f}_\nu^\star$ (possibly with different $(\boldsymbol{P}, \boldsymbol{D}, \boldsymbol{c})$). Thus $\mathbf{z}_\nu$ is component-wise identifiable through $f^\star$ in the sense of Defn. 2.2.*

*Proof.* Proof can be found in App. B.2. $\qquad\qquad\qquad\qquad\qquad\qquad\qquad\qquad\qquad\qquad\qquad\square$

*Remark 1.* Thm. 3.1 guarantees recovery of $\mathbf{z}_\iota$ up to an invertible block reparameterization and of $\mathbf{z}_\nu$ up to per-coordinate affine transformations by maximizing Eq. (4). In this context of single-cell perturbation prediction, these guarantees ensure that the perturbation-responsive latent subspace $\mathbf{z}_\nu$ can be disentangled from the invariant latent subspace $\mathbf{z}_\iota$. As a result, the true causal effects of perturbations can be isolated from dominant background cellular transcriptional programs, preventing confounding and allowing reliable estimation of perturbation-induced responses.

*Remark 2.* The identifiability guarantees in Thm. 3.1 crucially rely on the objective in Eq. (4), which combines a likelihood term and an alignment term across $\mathbf{u}$. The likelihood captures perturbation-induced variation in $\mathbf{z}_\nu$, while the alignment ensures $\mathbf{z}_\iota$ remains invariant. This combination is the key theoretical motivation for our method, enabling disentanglement of perturbation effects from background programs.

## 4 APPROACH: CONTRASTIVE DAG VARIATIONAL AUTOENCODER

In this section, we translate our theoretical findings into a practical framework for single-cell perturbation prediction. Building on the theoretical guarantee that the latent variables $\mathbf{z}_\iota$ and $\mathbf{z}_\nu$ can be recovered under the objective in Eq. (4), we introduce the *Contrastive DAG Variational Autoencoder* (cDAG-VAE), which detail how this objective can be implemented in practice, including the *Likelihood* term (Sec. 4.1) and the *Alignment* term (Sec. 4.2) in Eq. (4).

### 4.1 VARIATIONAL INFERENCE OF THE LIKELIHOOD TERM

Generally speaking, maximizing the likelihood term in Eq. (4) is intractable, as it involves integration in a high-dimensional space. Conventional approaches that resort to sum-product belief

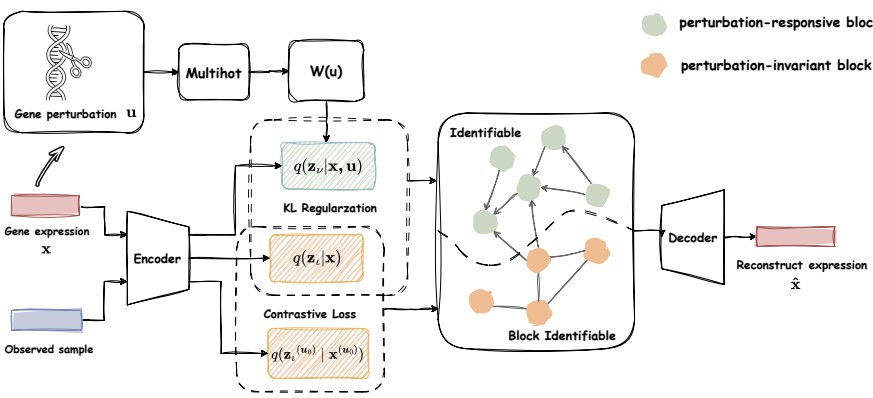

Figure 2: Framework of the proposed CDAG-VAE. *Perturbed cell expression profiles* $\mathbf{x}$ are used to learn the perturbation-responsive block $\mathbf{z}_\nu$, capturing the effects of perturbations indexed by $\mathbf{u}$. In parallel, *unperturbed control samples* $\mathbf{x}^{(\boldsymbol{u}_0)}$ are used for contrastive alignment of the perturbation-invariant block $\mathbf{z}_\iota$, ensuring that invariant cellular programs are disentangled from perturbation.

propagation or sampling algorithms often face with high computational cost (Bishop & Nasrabadi, 2006). To reduce the computational burden, we use a variational inference (Jordan et al., 1998; Blei et al., 2017), as follows:

$$\mathcal{L}_{\text{ELBO}} = \mathbb{E}_{q_\theta(\mathbf{z}_\nu, \mathbf{z}_\iota \mid \mathbf{x}, \mathbf{u})} \big[ \log p_\phi(\mathbf{x} \mid \mathbf{z}_\nu, \mathbf{z}_\iota, \mathbf{u}) \big] - D_{\text{KL}}(q_\theta(\mathbf{z}_\nu, \mathbf{z}_\iota \mid \mathbf{x}, \mathbf{u}) \,\|\, p_\phi(\mathbf{z}_\nu, \mathbf{z}_\iota \mid \mathbf{u})). \quad (5)$$

Here, $p_\phi(\mathbf{z}_\nu, \mathbf{z}_\iota \mid \mathbf{u})$ denotes the prior distribution arising from assumptions on the latent space, $q_\theta(\mathbf{z}_\nu, \mathbf{z}_\iota \mid \mathbf{x}, \mathbf{u})$ denotes a variational posterior approximating the true posterior $p_\phi(\mathbf{z}_\nu, \mathbf{z}_\iota \mid \mathbf{x}, \mathbf{u})$, and $D_{\text{KL}}$ denotes the KL divergence. Specifically, based on our model assumptions in Eqs. 1 and 2, the prior distribution can be factorized as follows:

$$p_\phi(\mathbf{z}_\nu, \mathbf{z}_\iota \mid \mathbf{u}) = p_\phi(\mathbf{z}_\nu \mid \mathbf{u}, \mathbf{z}_\iota) \, p_\phi(\mathbf{z}_\iota), \quad (6)$$

For the variational posterior, our goal is not only to recover $\mathbf{z}_\iota$ up to an invertible block reparameterization and $\mathbf{z}_\nu$ up to permutation, as discussed in Thm. 3.1, but more importantly, to learn the causal structure over $\mathbf{z}_\nu$, since it encodes perturbation information that is central to single-cell perturbation prediction. Therefore, we consider the following structured variational posterior:

$$q_\theta(\mathbf{z}_\nu, \mathbf{z}_\iota \mid \mathbf{x}, \mathbf{u}) = q_\theta(\mathbf{z}_\nu \mid \mathbf{x}, \mathbf{u}) \, q_\theta(\mathbf{z}_\iota \mid \mathbf{x}). \quad (7)$$

We here employ variational inference with a structured posterior that factorizes as in Eq. 7. This factorization preserves the internal structures of $\mathbf{z}_\nu$ and $\mathbf{z}_\iota$ while ignoring their mutual dependencies, thereby balancing computational efficiency with the ability to capture meaningful latent factors. Such a design also facilitates subsequent learning of causal structures and perturbation effects.

### 4.2 LEARNING UNPERTURBED EFFECT VIA THE ALIGNMENT TERM

The alignment term in the objective in Eq. (4), as formalized in Thm. 3.1, is a key component that distinguishes this work from previous approaches. Although the likelihood term in Eq. 5 attempts to capture the invariant block $\mathbf{z}_\iota$, our theoretical findings in Thm. 3.1 show that proper identification of $\mathbf{z}_\iota$ fundamentally requires the presence of the alignment term. In other words, without this contrastive alignment across perturbation conditions, $\mathbf{z}_\iota$ cannot be reliably disentangled from the perturbation-responsive block $\mathbf{z}_\nu$. Essentially, the alignment term can theoretically recover $\mathbf{z}_\iota$ through the loss $\|f_\iota(\mathbf{x}^{(\boldsymbol{u}_0)}) - f_\iota(\mathbf{x})\|_2^2$, as defined in Eq. 4, which exploits the property that $\mathbf{z}_\iota$ is invariant across perturbation conditions $\mathbf{u}$. This invariance can also be observed in Fig. 1a. Consequently, the alignment term can be implemented by directly enforcing invariance on $\mathbf{z}_\iota$ across $\mathbf{u}$, as follows:

$$\mathcal{L}_{\text{contrast}}(\mathbf{x}, \mathbf{x}^{(\boldsymbol{u}_0)}) = \|\mathbf{z}_\iota - \mathbf{z}_\iota^{(\boldsymbol{u}_0)}\|_2^2. \quad (8)$$

We emphases that the alignment term, implemented by Eq. 8, is crucial, as it ensures that information contained in $\mathbf{z}_\iota$ is not inadvertently absorbed by $\mathbf{z}_\nu$. In other words, if $\mathbf{z}_\iota$ cannot be properly

identified, information pertaining to $\mathbf{z}_\iota$ may leak into $\mathbf{z}_\nu$. In such a scenario, the causal relationships among the components of $\mathbf{z}_\iota$ cannot be reliably learned, since the invariant information is contaminated by the perturbation-responsive block. In our CDAG-VAE, we model the variational posteriors $q_\theta(\mathbf{z}_\nu \mid \mathbf{x}, \mathbf{u})$ and $q_\theta(\mathbf{z}_\iota \mid \mathbf{x})$ as multivariate normal distributions, and instantiate $f_\nu$ and $f_\iota$ by their corresponding posterior means.

## 4.3 THE PROPOSED CONTRASTIVE DAG VARIATIONAL ANTOENCODER

Building on the variational inference framework and the alignment principle across perturbation conditions $\mathbf{u}$ above, we define the overall objective function for CDAG-VAE as a combination of the likelihood-based ELBO and the contrastive alignment loss, according to Thm. 3.1.

$$\mathcal{L}_{\theta,\phi} = \mathbb{E}_{(\mathbf{x},\mathbf{u}) \sim p_{\phi^\circ}(\mathbf{x}|\mathbf{u})} \left[ \|\mathbf{x} - \hat{\mathbf{x}}\|_2^2 + \beta_\nu \, \mathcal{L}_{\text{KL-v}}(\mathbf{x},\mathbf{u}) + \beta_\iota \, \mathcal{L}_{\text{KL-i}}(\mathbf{x}) + \alpha \, \mathcal{L}_{\text{contrast}}(\mathbf{x}, \mathbf{x}^{(\mathbf{u}_0)}) \right]. \quad (9)$$

where $\hat{\mathbf{x}}$ denotes the reconstruction of $\mathbf{x}$, $\mathcal{L}_{\text{KL-v}}(\mathbf{x},\mathbf{u}) = D_{\text{KL}}(q_\theta(\mathbf{z}_\nu \mid \mathbf{x}, \mathbf{u}) \| p_\phi(\mathbf{z}_\nu \mid \mathbf{u}, \mathbf{z}_\iota))$, $\mathcal{L}_{\text{KL-i}}(\mathbf{x}) = D_{\text{KL}}(q_\theta(\mathbf{z}_\iota \mid \mathbf{x}) \| p_\phi(\mathbf{z}_\iota))$, $\alpha$ is the weighting hyperparameter motivated from Thm. 3.1, and for each $\mathbf{x}$, $\mathbf{x}^{(\mathbf{u}_0)}$ is a paired observation randomly sampled from $p_{\phi^\circ}(\mathbf{x}|\mathbf{u}_0))$. We here introduce $\beta_\nu, \beta_\iota$ motivate by Higgins et al. (2017) to balance the contributions of the KL terms.

In summary, the overall objective in Eq. 9 balances multiple goals: the reconstruction term ensures that the latent representations retain sufficient information from the original data, the KL term for $\mathbf{z}_\nu$ encourages encoding of perturbation-specific effects, the KL term for $\mathbf{z}_\iota$ regulates the invariant block, and the contrastive alignment term ensures that perturbation-invariant information is disentangled from perturbation-specific variation. Together, these components allow CDAG-VAE to recover meaningful latent factors while disentangling perturbation effects from invariant cellular programs.

## 5 EMPERICAL FINDINGS

**Numerical Simulation.** We first conduct simulations to verify our theoretical results under idealized assumptions. To this end, we generate synthetic data according to our latent causal generative model in Eqs. 1- 3. More details can be found in App. C.2. This setup allows us to systematically assess the recovery of latent subspaces and causal structures under controlled conditions. For evaluation, following Sorrenson et al. (2020); Khemakhem et al. (2020), we use the mean correlation coefficient (MCC) to quantify component-wise recovery of $\mathbf{z}_\nu$. Specifically, MCC measures the correlation between each learned component of $\mathbf{z}_\nu$ and its corresponding ground-truth component, with a value of 1 indicating perfect recovery. For block-wise evaluation of $\mathbf{z}_\iota$, we report the kernel regression $R^2$, following Von Kügelgen et al. (2021), which captures the nonlinear relationship between the learned block and its ground-truth counterpart. Values closer to 1 indicate better block-level disentanglement.

Table 1 shows that the contrastive alignment term substantially improves identifiability. For the variant block $\mathbf{z}_\nu$, MCC increases from 0.81 to 0.86 and block-wise $R^2$ from 0.93 to 0.95, indicating more accurate recovery of intervention-specific factors. The effect is even more pronounced for the invariant block $\mathbf{z}_\iota$, whose $R^2$ rises from 0.66 to 0.97, highlighting the crucial role of con-

Table 1: Results on simulation data.

| Contrastive Alignment | MCC | $R^2$ (nonlinear) | |
| --- | --- | --- | --- |
| | Var. $\mathbf{z}_\nu$ (identifiable) | Var. $\mathbf{z}_\nu$ (block-identifiable) | Inv. $\mathbf{z}_\iota$ (block-identifiable) |
| ✗ | $0.81_{\pm 0.0306}$ | $0.93_{\pm 0.0120}$ | $0.66_{\pm 0.0281}$ |
| ✓ | $\mathbf{0.86}_{\pm 0.0285}$ | $\mathbf{0.95}_{\pm 0.0020}$ | $\mathbf{0.97}_{\pm 0.0077}$ |

trastive alignment in disentangling invariant programs from perturbation-induced effects. These results confirm our theoretical claims: contrastive alignment enhances recovery of $\mathbf{z}_\iota$ and prevents its information from being absorbed into $\mathbf{z}_\nu$, thereby facilitating both accurate the component-wise and block-identifiability guarantees in Thm. 3.1.

**Real-world Perturbation** For real-world perturbation data, we consider the large-scale Perturbseq dataset from (Norman et al., 2019), referred to as Norman2019. It comprises 105,528 cells from an erythrocytic leukemia cell line (K562) subjected to CRISPR activation (Gilbert et al., 2014) targeting 112 genes, resulting in 105 single-gene and 131 double-gene perturbations. The regulatory effect on each target gene's expression can be modeled as a intervention (Zhang et al., 2023). Each

perturbation condition contains between 50 and 2,000 cells. Across all conditions, each cell is represented as a 5,000-dimensional vector $\mathbf{x}$, corresponding to the gene expression levels.

EXPERIMENTAL SETUP. We partition the Norman2019 dataset into training and testing splits as follows. The training set consists of all unperturbed cells together with the 105 single-gene perturbation datasets $\mathcal{X}_1, \ldots, \mathcal{X}_{105}$. For each single-gene dataset with more than 800 cells, we randomly hold out 96 cells to form a *single-gene test set*, while the remaining cells are included in training. The *double-gene test set* comprises the 112 datasets $\mathcal{X}_{106}, \ldots, \mathcal{X}_{217}$, which are entirely reserved for evaluation and never used during training. This setup ensures that the model is trained on existing perturbations, but is evaluated on both held-out single-gene cells and, more importantly, on unseen combinatorial perturbations. In addition, for the differentially expressed (DE) gene–focused analysis in App. C.6, we construct a complementary 20-dimensional version of the Norman2019 dataset, where each cell is represented by its expression over the top 20 most DE genes.

A key architectural choice in CDAG-VAE is how capacity is allocated between the variant and invariant subspaces. We assign the invariant subspace substantially more latent dimensions than the variant subspace, reflecting its role in modeling complex background programs [5]. To test sensitivity, we vary the total latent dimension across $\{10, 35, 70, 105\}$, scaling both subspaces proportionally, and evaluate the effect on reconstruction fidelity and disentanglement. We benchmark CDAG-VAE against three representative baselines, Discrepancy-VAE (Zhang et al., 2023), SENA-discrepancy-VAE (SENA) (de la Fuente et al., 2025), sVAE+ (Lopez et al., 2022), SAMS-VAE (Bereket & Karaletsos, 2023) reporting results averaged over five random seeds for each model. We also implement a variant of the proposed CDAG-VAE, namely DAG-VAE, which excludes the contrastive alignment term. All results correspond to the final trained model, with extended evaluations and ablation studies provided in App. C.5.

SINGLE-GENE PERTURBATION. To evaluate the generative capacity of our model on perturbation types, we focus on the 14 single-gene conditions with more than 800 available cells. For each such condition, we generate 96 synthetic cells from the learned model and compare them against 96 held-out real cells that were not used during training. Evaluation is conducted using $R^2$ [6] across all genes. Our model demonstrates high fidelity, with an average $R^2$ of 0.99 across the 14 conditions. This result confirms that the proposed latent-variable formulation can faithfully reproduce cellular responses for known perturbations, even on held-out samples not seen during training. Complementing the $R^2$ results, we further report the root mean squared error (RMSE), which quantifies absolute deviations in predicted expression levels. Consistently low RMSE values demonstrate that CDAG-VAE not only explains variance but also faithfully captures absolute gene expression magnitudes, an essential requirement for biological interpretability. Intriguingly, when we developed a CDAG-VAE variant incorporating an MMD loss to explicitly model higher-order statistics such as variance and covariance, its RMSE slightly increased compared to our original model, while still comprehensively outperforming all baselines. This suggests a potential trade-off between achieving the lowest error in mean expression and faithfully capturing the full distributional complexity of cellular populations. See App. C.5 for more experimental results for single-gene perturbations.

DOUBLE-GENE PERTURBATION. Building upon single-gene perturbations, we next subjected our model to a far more stringent test: out-of-distribution generalization to 112 unseen double-gene perturbations. This task constitutes a true zero-shot prediction challenge, as no cells from these combinatorial interventions were seen during training. To evaluate performance, we again compared the population-average expression profile of generated cells against that of the held-out real cells. Despite this challenge, our model achieves strong performance, with $R^2$ of 0.98 across all measured genes, as shown in Figure 3. These results indicate that the model successfully composes knowledge from

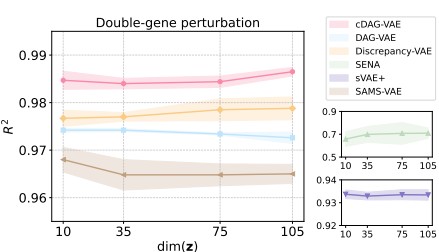

Figure 3: $R^2$ on double-gene perturbation

---

[5]See App. C.5 for an extended ablation study on the effect of allocating latent capacity between $\mathbf{z}_\nu$ and $\mathbf{z}_\iota$.

[6]On real data, $R^2$ is computed at the population-average level: we compare the mean predicted expression per perturbation to the mean observed expression of the corresponding cells. In simulations, $R^2$ is computed against the ground truth (cell-wise or after optimal nonlinear alignment). See App. B.4 for details.

Table 2: RMSE on Double-gene perturbations prediction.

| Method | Latent dimension | | | |
|---|---|---|---|---|
| | 10 | 35 | 75 | 105 |
| **Discrepancy-VAE** (Zhang et al., 2023) | $0.6084_{\pm 0.0045}$ | $0.6037_{\pm 0.0025}$ | $0.6075_{\pm 0.0072}$ | $0.6082_{\pm 0.0045}$ |
| **SENA** (de la Fuente et al., 2025) | $0.8573_{\pm 0.0205}$ | $0.8514_{\pm 0.0248}$ | $0.8507_{\pm 0.0396}$ | $0.8483_{\pm 0.0248}$ |
| **sVAE+** (Lopez et al., 2022) | $0.5663_{\pm 0.0009}$ | $0.5667_{\pm 0.0008}$ | $0.5665_{\pm 0.0011}$ | $0.5664_{\pm 0.0012}$ |
| **SAMS-VAE** (Bereket & Karaletsos, 2023) | $0.4605_{\pm 0.0020}$ | $0.4631_{\pm 0.0024}$ | $0.4632_{\pm 0.0017}$ | $0.4629_{\pm 0.0014}$ |
| **DAG-VAE (Ours)** | $0.4557_{\pm 0.0005}$ | $0.4563_{\pm 0.0005}$ | $0.4577_{\pm 0.0005}$ | $0.4623_{\pm 0.0041}$ |
| **cDAG-VAE (Ours)** | $0.4493_{\pm 0.0019}$ | $0.4494_{\pm 0.0008}$ | $0.4489_{\pm 0.0009}$ | $0.4474_{\pm 0.0007}$ |

single-gene interventions to predict the transcriptional consequences of unseen combinatorial perturbations, highlighting its ability to capture causal structure rather than merely memorizing training distributions. Complementing these results, we also evaluate the RMSE to quantify absolute prediction accuracy under out-of-distribution conditions, as shown in Table 2. Consistently low RMSE values indicate that cDAG-VAE not only generalizes the relative variance structure captured by $R^2$ but also preserves absolute gene-expression magnitudes in unseen double-gene perturbations. This robustness underscores the model's ability to extrapolate causal effects beyond the training distribution. Beyond VAE-based baselines, we also compare cDAG-VAE to non-generative predictors: a classical additive linear model and the GEARS architecture for combinatorial perturbation prediction. As detailed in App. C.7, we show a perspective on latent causal model for double-gene perturbation.

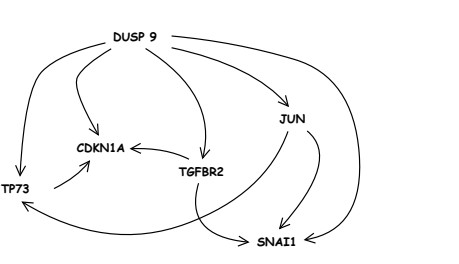

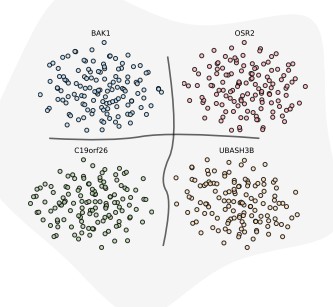

(a) Learned causal structure ($\mathbf{z}_\nu$).        (b) Perturbation-invariant representation ($\mathbf{z}_\iota$).

Figure 4: Illustration of the learned latent space. (a) The DAG structure over the variant subspace $\mathbf{z}_\nu$. (b) Two-dimensional visualization of the estimated invariant subspace $\mathbf{z}_\iota$.

STRUCTURE LEARNING. Following (Zhang et al., 2023), we evaluated the DAG structure, which corresponds to a learned coarse-grained gene regulatory network between the learned programs of the target genes by hard assignment via maximal intervention effect, obtained by the proposed cDAG-VAE. The DAG Fig. 4a demonstrates high biological fidelity by recapitulating key known regulatory interactions. These include the TGFBR2→SNAI1 axis essential for epithelial–mesenchymal transition (EMT) (Vincent et al., 2009; Fan et al., 2025), the canonical TP73→CDKN1A tumor suppressor pathway governing cell-cycle arrest (Schmidt et al., 2021), and the DUSP9-mediated inhibition of JUN, a critical negative feedback mechanism in MAPK signaling (Emanuelli et al., 2008). This recovery of established biological mechanisms validates the utility of our approach for causal discovery from single-cell data. Full mechanistic notes for all program-level edges are provided in App. C.3.

UNPERTURBED LATENT SPACE. For the invariant block $\mathbf{z}_\iota$, we systematically evaluated whether its representation remains stable across perturbations. by examining all single-gene conditions in the test set. As shown in Fig. 4b, a t-SNE projections (Maaten & Hinton, 2008) for four representative perturbations, where cells from distinct perturbations remain intermixed rather than forming separate

clusters. This indicates that perturbation identity does not explain variation in the invariant block, and demonstrates that $\mathbf{z}_\iota$ captures background cellular programs that generalize beyond training conditions. See App. C.4 for more details.

## 6  CONCLUSION.

In this work, we introduce CDAG-VAE, a contrastive variational framework that decomposes single-cell variation into perturbation-responsive (variant) factors and invariant background programs. Under the assumptions stated in this work, we provide block-identifiability guarantees for the variant and invariant components and further show that the variant subspace itself is identifiable, thereby offering theoretical support for reliable causal discovery under sparse interventions. Empirically, on synthetic data and large-scale single-cell perturbation benchmarks, CDAG-VAE recovers biologically interpretable programs and consistently improves out-of-distribution prediction on unseen double-gene combinations over strong baselines. Together, these results establish a theoretically grounded and empirically validated route toward data-efficient in-silico prioritization of combinatorial interventions.

**Ethics Statement.** We confirm that this study complies with the ethical standards of ICLR, with no involvement of private or sensitive information.

**Reproducibility statement.** We have taken extensive measures to ensure the reproducibility of our work. Appendix C.1 presents the pseudocode of our method, while Appendix C.2 describes the data generation procedure for simulation experiments along with the corresponding training setup and hyperparameter configurations. For experiments on real datasets, detailed hyperparameter choices are included in Appendix C.5.

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

# Identifying Unperturbed Cellular Programs Enables Accurate Single-Cell Perturbation Prediction

## Appendices

### CONTENTS

We organize the Appendix as follows.

# A RELATED WORK

**Disentangling single-cell perturbation effects.** A central challenge in single-cell perturbation modeling is to separate intervention effects from intrinsic cellular variability. Deep generative approaches have shown strong performance on this task. scGen (Lotfollahi et al., 2019) models perturbations as additive shifts in a latent space, while CPA (Lotfollahi et al., 2023) factorizes each cell into basal state and perturbation effect. chemCPA (Hetzel et al., 2022) extends CPA with chemical structure embeddings and dosage information, enabling zero-shot predictions for unseen compounds. Other methods incorporate biological priors or contrastive objectives: GEARS (Roohani et al., 2024) uses gene-gene interaction graphs for improved generalization across perturbation combinations, and contrastive VAEs have been applied in optical pooled screening to disentangle stable identity from perturbation-driven variation (Wang et al., 2023). Despite empirical successes, most of these models treat disentanglement statistically rather than causally, which limits interpretability. Recent work has incorporated sparsity into latent-variable models to encourage identifiable and interpretable representations. CausCell (Gao et al., 2025b) enables counterfactual generation via SCM-guided diffusion, but critically depends on a predefined causal graph, limiting its applicability when causal structures are unknown or hard to specify. sVAE+ (Lopez et al., 2022), SAMS-VAE (Bereket & Karaletsos, 2023), scShift (Dong et al., 2024) impose sparse structure or mechanism shifts in the latent space to model perturbation-induced variation, scShift learns flat latent embeddings and performs causal discovery only post hoc, without an end-to-end structural causal model for composing unseen combinatorial perturbations. Recent advances such as discrepancy-VAE (Zhang et al., 2023), and its interpretable variant (de la Fuente et al., 2025) align latent-variable models with identifiable causal semantics, pointing toward representations that are both intervention-sensitive and explanatory. Building on these advances, our approach moves beyond purely statistical factorization, ensuring that the learned representations reflect genuine causal effects of perturbations.

**Identifiable causal representations.** A key aim in modeling complex systems is to learn low-dimensional latent variables $\mathbf{z}$ from high-dimensional data $\mathbf{x}$ that match the true generative factors (independent components) (Hyvärinen et al., 2001). Nonlinear ICA showed that such components are not identifiable from i.i.d. data without extra assumptions (Hyvärinen & Pajunen, 1999). Identifiable variants address this by introducing an auxiliary variable $\mathbf{u}$ so that latent factors $\{z_i\}_{i=1}^p$ are conditionally independent given $\mathbf{u}$ (Hyvarinen & Morioka, 2016; 2017). The iVAE framework (Khemakhem et al., 2020), built on VAEs (Kingma et al., 2013; Rezende et al., 2014), proves identifiability of both $\mathbf{z}$ and $p(\mathbf{x} \mid \mathbf{z})$ under mild conditions. Recent approaches impose structure in latent space: DAG-based models enforce acyclicity (Lippe et al., 2022; Liu et al., 2022; 2024; Ahuja et al., 2023), while factorized designs split latent variables into invariant, intervention-specific, and interaction parts (Von Kügelgen et al., 2021; Kong et al., 2022; Gao et al., 2025a). While prior methods establish identifiability via auxiliary conditioning or broad structural constraints, our model ties perturbations directly to latent mechanisms. This design moves beyond heuristic augmentations or globally factorized latents, making our framework specifically tailored to single-cell perturbation.

**Contrastive representation learning.** Contrastive multi-view learning learns invariances across views or modalities (e.g., SimCLR, BYOL, CLIP-style training) but typically relies on heuristic augmentations whose invariants need not align with causal structure (Chen et al., 2020; Grill et al., 2020; Radford et al., 2021; Cai et al., 2024; 2025; Tschannen et al., 2020; Von Kügelgen et al., 2021). Aliee et al. (2023) learn conditionally invariant representations by leveraging variability across observational environments (patients, batches, platforms) to suppress domain-specific artefacts while preserving biological signal. In single-cell analysis, Weinberger et al. (2023) contrast background and target datasets—extending to multi-omics—to isolate salient structure, but provide no identifiability guarantees. For perturbation screens, supervised contrastive VAEs use guide labels with HSIC to isolate perturbation effects from background heterogeneity (Tu et al., 2024). Concurrently, Mao et al. (2024) posit a three-way factorization (covariate, treatment, interaction) and promote independence via structural constraints and adversarial training; while principled, this fixed design may underfit non-classical responses and its identifiability hinges on stringent experimental designs. Unlike contrastive or domain-invariant models, we obtain block identifiability for the perturbation-invariant block and component-wise identifiability for the perturbation-responsive block under a weight-variant latent SCM, thereby performing CRL in the latent space and recovering the latent causal graph among responsive variables.

# B PROOFS AND TECHNICAL DETAILS

## B.1 NOTATION

Random vectors are denoted by bold lowercase letters (e.g., $\mathbf{a}$), with their realizations written as bold symbols (e.g., $\boldsymbol{a}$). Matrix-valued random variables are denoted by bold uppercase letters (e.g., $\mathbf{A}$), with realizations $\boldsymbol{A}$. Scalar random variables are denoted by serif letters (e.g., $\mathrm{a}$), with realizations written as plain letters (e.g., $a$). A complete list of the notations employed throughout this paper is provided below:

Table 3: Complete notation used in §2–4.

| Spaces | |
| --- | --- |
| $\mathcal{X} \subseteq \mathbb{R}^{d_x}$ | Gene expression space (observations). |
| $\mathcal{U}$ | Space of perturbation identities/environments (e.g., one-hot). |
| $\mathcal{Z}_\iota \subseteq \mathbb{R}^{d_\iota}$ | Invariant latent subspace. |
| $\mathcal{Z}_\nu \subseteq \mathbb{R}^{d_\nu}$ | Variant/perturbation-responsive latent subspace. |
| $\mathcal{Z} = \mathcal{Z}_\iota \times \mathcal{Z}_\nu$ | Full latent space; $d_z = d_\iota + d_\nu$. |
| $\mathcal{N}_\iota \subseteq \mathbb{R}^{d_\iota},\ \mathcal{N}_\nu \subseteq \mathbb{R}^{d_\nu}$ | Supports of exogenous noises for $\mathbf{z}_\iota$ and $\mathbf{z}_\nu$. |
| **Random variables and their realizations** | |
| $\mathbf{x}^{(\boldsymbol{u}_0)} \in \mathcal{X}$ | Control/anchor expression under $\boldsymbol{u}_0$; realization $\boldsymbol{x}^{(\boldsymbol{u}_0)}$. |
| $\mathbf{x} \in \mathcal{X}$ | Perturbed expression under $\mathbf{u} \neq \boldsymbol{u}_0$; realization $\boldsymbol{x}$. |
| $\mathbf{z}_\iota \in \mathcal{Z}_\iota$ | Invariant latent variables; realization $\boldsymbol{z}_\iota$. |
| $\mathbf{z}_\nu \in \mathcal{Z}_\nu$ | Variant latent variables; realization $\boldsymbol{z}_\nu$. |
| $\mathbf{z} = (\mathbf{z}_\iota, \mathbf{z}_\nu) \in \mathcal{Z}$ | All latent variables; realization $\boldsymbol{z} = (\boldsymbol{z}_\iota, \boldsymbol{z}_\nu)$. |
| $\mathbf{z}_\nu^{(\boldsymbol{u}_0)}, \mathbf{z}_\nu$ | Variant latents under control $\boldsymbol{u}_0$ and perturbation $\mathbf{u}$ (realizations $\boldsymbol{z}_\nu^{(\boldsymbol{u}_0)}, \boldsymbol{z}_\nu$). |
| $\tilde{\mathbf{z}}_\nu$ | Variant latents of the paired sample in contrastive protocol (realization $\tilde{\boldsymbol{z}}_\nu$). |
| $\mathrm{z}_{\nu,i}$ | $i$-th coordinate of $\mathbf{z}_\nu$ (realization $z_{\nu,i}$; similarly $\tilde{z}_{\nu,i}$ for $\tilde{\mathbf{z}}_\nu$). |
| $\mathbf{n}_\iota \in \mathcal{N}_\iota, \mathbf{n}_\nu \in \mathcal{N}_\nu$ | Exogenous noises; realizations $\boldsymbol{n}_\iota, \boldsymbol{n}_\nu$. |
| **Maps and mechanisms** | |
| $g : \mathcal{Z} \to \mathcal{X}$ | Generative map producing $\mathbf{x}$ from $\mathbf{z}$; assumed diffeomorphic. |
| $g_z : \mathcal{U} \times \mathcal{N}_\nu \to \mathcal{Z}_\nu$ | Abstract latent causal mechanism for $\mathbf{z}_\nu$. |
| $f = (f_\iota, f_\nu) : \mathcal{X} \to \mathbb{R}^{d_\iota} \times \mathbb{R}^{d_\nu}$ | Inference encoders / learned codes (realizations $\boldsymbol{f}$ evaluated at $\boldsymbol{x}$). |
| **Latent SCM parameters (weight-variant)** | |
| $\boldsymbol{\lambda}_{\iota\iota}, \boldsymbol{\lambda}_{\nu\iota}(\mathbf{u}), \boldsymbol{\lambda}_{\nu\nu}(\mathbf{u})$ | Block weight matrices (strictly upper triangular; order $\mathbf{z}_\iota \prec \mathbf{z}_\nu$). Realizations $\boldsymbol{\Lambda}_{..}(\boldsymbol{u})$. |
| $\boldsymbol{\mu}_\iota, \boldsymbol{\beta}_\iota;\ \boldsymbol{\mu}_\nu(\mathbf{u}), \boldsymbol{\beta}_\nu(\mathbf{u})$ | Gaussian noise means and variances for $\mathbf{n}_\iota,\ \mathbf{n}_\nu$ (environment-dependent for $\nu$). Realizations $\boldsymbol{m}_{.}, \boldsymbol{b}_{.}$. |
| $\tau_j(\mathbf{u}) = \beta_{\nu,j}^{-1}(\mathbf{u})$ 
 $\kappa_j(\mathbf{u}) = \tau_j(\mathbf{u})\mu_{\nu,j}(\mathbf{u})$ | Precision and natural mean for the $j$-th $\nu$-noise (used in richness/coverage assumptions). |
| **Objectives and losses** | |
| $\mathcal{L}_{\text{ELBO}}$ | Evidence lower bound. |
| $\mathcal{L}_{\text{contrast}}$ | Contrastive alignment loss on $f_\iota(\mathbf{x}^{(\boldsymbol{u}_0)})$ and $f_\iota(\mathbf{x})$. |
| $\mathcal{J}_{\text{obj}}$ | Joint objective likelihood minus alignment term. |
| $\mathcal{L}_{\theta,\phi}$ | Total loss function combining all objectives. |

### B.2 PROOF OF THM. 3.1

Before proving, we first restate the theorem for clarity:

**Theorem 3.1** (Identifiability of the proposed latent causal generative model). *Consider smooth inference encoders $f : \mathcal{X} \to \mathbb{R}^{d_\iota + d_\nu}$, decomposed as $f(\mathbf{x}) = (f_\iota(\mathbf{x}), f_\nu(\mathbf{x}))$ with $\dim(f_\iota) = d_\iota$ and $\dim(f_\nu) = d_\nu$. Suppose Asms. 3.1 to 3.3 hold. Define the joint objective*

$$\mathcal{J}_{obj}(\phi, f) := \underbrace{\mathbb{E}_{(\mathbf{x},\mathbf{u}) \sim p_{\phi^\circ}}[\log p_\phi(\mathbf{x} \mid \mathbf{u})]}_{Likelihood} - \alpha \underbrace{\mathbb{E}_{(\mathbf{x}^{(\boldsymbol{u}_0)}, \mathbf{x})}\left[\|f_\iota(\mathbf{x}^{(\boldsymbol{u}_0)}) - f_\iota(\mathbf{x})\|_2^2\right]}_{Alignment\ across\ \mathbf{u}}, \qquad (4)$$

*where $\alpha > 0$ is a scaling constant, $(\mathbf{x}^{(\boldsymbol{u}_0)}, \mathbf{x})$ are positive pairs following Asm. 3.5, and $\mathbf{u} \sim q_{\mathbf{u}}$ as in Asm. 3.4. Let $(\phi^\star, f^\star)$ be a global maximizer of Eq. (4). At the global maximizer, the optimization is constrained so that for any $\mathbf{z}_\nu \in \mathcal{Z}_\nu$, the map $\mathbf{z}_\iota \mapsto f_\iota^\star \circ g(\mathbf{z})$ is injective.*

*Then, for any two global maximizers $(\phi^\star, f^\star)$ and $(\tilde{\phi}^\star, \tilde{f}^\star)$ that realize the true marginal $p_{\phi^\circ}(\mathbf{x}|\mathbf{u})$, i.e., $\mathbb{E}[\log p_{\tilde{\phi}^\star}] = \mathbb{E}[\log p_{\phi^\star}] = \mathbb{E}[\log p_{\phi^\circ}]$, the corresponding encodings satisfy:*

1. (Block-identifiability of $\mathbf{z}_\iota$). *There exist bijections $h_\iota, \tilde{h}_\iota : \mathcal{Z}_\iota \to \mathbb{R}^{d_\iota}$ such that $f_\iota^\star(\mathbf{x}) = h_\iota(\mathbf{z}_\iota)$ and $\tilde{f}_\iota^\star(\mathbf{x}) = \tilde{h}_\iota(\mathbf{z}_\iota)$ a.s., thus $\mathbf{z}_\iota$ is block-identifiable through $f^\star$ in the sense of Defn. 2.1.*

2. (Component-wise identifiability of $\mathbf{z}_\nu$). *There exist permutation $\boldsymbol{P} \in \mathbb{R}^{d_\nu \times d_\nu}$, diagonal $\boldsymbol{D} \succ 0$, and $\boldsymbol{c} \in \mathbb{R}^{d_\nu}$ such that $f_\nu^\star(\mathbf{x}) = \boldsymbol{P}\boldsymbol{D}\mathbf{z}_\nu + \boldsymbol{c}$ a.s.; likewise for $\tilde{f}_\nu^\star$ (possibly with different $(\boldsymbol{P}, \boldsymbol{D}, \boldsymbol{c})$). Thus $\mathbf{z}_\nu$ is component-wise identifiable through $f^\star$ in the sense of Defn. 2.2.*

*Proof.* We first decompose the learning objective into two terms:

$$\mathcal{J}_{obj}(\phi, f) := \underbrace{\mathbb{E}_{(\mathbf{x},\mathbf{u})}[\log p_\phi(\mathbf{x}|\mathbf{u})]}_{Term\ I} - \alpha \underbrace{\mathbb{E}_{(\mathbf{x}^{(\boldsymbol{u}_0)}, \mathbf{x})}\left[\|f_\iota(\mathbf{x}^{(\boldsymbol{u}_0)}) - f_\iota(\mathbf{x})\|_2^2\right]}_{Term\ II}, \quad \alpha > 0, \qquad (10)$$

Now, we construct the proof in the following two steps:

**Step 1 ($\mathbf{z}_\iota$ is block-identifiable).** Term I depends only on $\phi$, not on a specific $f$. At any likelihood-optimal $\phi$ realizing $p_{\phi^\circ}(\mathbf{x}|\mathbf{u})$, we may analyze encoders via the true diffeomorphism $g$ from Asm. 3.2. Set

$$h := f \circ g : \mathcal{Z} \to \mathbb{R}^{d_\iota + d_\nu}.$$

Since the true generative mapping $g$ is diffeomorphic and the inference encoder $f$ is smooth, we have $h$ is $C^1$ with respect to the latent measure.

*(a) The infimum of Term II is 0 and is attained at a global maximizer.* By Asm. 3.5, positive pairs satisfy $\mathbf{z}_\iota^{(\boldsymbol{u}_0)} = \mathbf{z}_\iota$ a.s. Consider encoders whose invariant part depends only on the invariant latents, i.e., choose $h_\iota(\mathbf{z}) = \psi(\mathbf{z}_\iota)$ with some measurable $\psi$, and let $h_\nu$ be arbitrary. Then for any positive pair, $\|h_\iota(\mathbf{z}^{(\boldsymbol{u}_0)}) - h_\iota(\mathbf{z})\|_2 = \|\psi(\mathbf{z}_\iota^{(\boldsymbol{u}_0)}) - \psi(\mathbf{z}_\iota)\|_2 = 0$ a.s., so the infimum of Term II is 0 and is achieved by such $h$. Since $g$ is invertible (onto its image), there exists an encoder $f = h \circ g^{-1}$ realizing this $h$ at the data level.

Moreover, Term I depends only on $\phi$ (not on the choice of $f$), so among all pairs $(\phi, f)$ that realize $p_{\phi^\circ}(\mathbf{x} \mid \mathbf{u})$, the objective is maximized by choosing $f$ that attains the infimum of Term II. Hence any global maximizer $(\phi^\star, f^\star)$ must satisfy

$$\mathbb{E}\left[\|f_\iota^\star(\mathbf{x}^{(\boldsymbol{u}_0)}) - f_\iota^\star(\mathbf{x})\|_2^2\right] = 0 \quad \implies \quad f_\iota^\star(\mathbf{x}^{(\boldsymbol{u}_0)}) = f_\iota^\star(\mathbf{x}) \text{ a.s.} \qquad (11)$$

*(b) Invariance along excited directions forces dependence only on $\mathbf{z}_\iota$.* Write $h^\star = f^\star \circ g = (h_\iota^\star, h_\nu^\star)$, where $h_\iota^\star := f_\iota^\star \circ g$ and $h_\nu^\star := f_\nu^\star \circ g$. From Eq. (11),

$$h_\iota^\star(\mathbf{z}_\iota, \mathbf{z}_\nu^{(\boldsymbol{u}_0)}) = h_\iota^\star(\mathbf{z}_\iota, \mathbf{z}_\nu) \quad \text{a.s.} \qquad (12)$$

By Asms. 3.4 and 3.5, for each $j \in [d_\nu]$ there is a set $\mathcal{U}_j \subseteq \mathcal{U} \setminus \{u_0\}$ with $q_{\mathbf{u}}(\mathcal{U}_j) > 0$ such that either the incoming weights $\boldsymbol{\lambda}_j(\mathbf{u})$ change or the univariate noise natural parameters $(\kappa_j(\mathbf{u}), \tau_j(\mathbf{u}))$ change relative to $u_0$. Under the acyclic order, the scalar equation for node $j$ reads

$$\mathbf{z}_{\nu,j} = \boldsymbol{\lambda}_j(\mathbf{u})^\top \mathbf{z}_{\mathrm{pa}(j)} + \mathbf{n}_{\nu,j}, \qquad \mathbf{n}_{\nu,j} \sim \mathcal{N}\big(\mu_{\nu,j}(\mathbf{u}), \beta_{\nu,j}(\mathbf{u})\big),$$

hence, conditional on $\mathbf{z}_{\mathrm{pa}(j)}$ and $\mathbf{u}$,

$$\mathbf{z}_{\nu,j} \mid \mathbf{z}_{\mathrm{pa}(j)}, \mathbf{u} \sim \mathcal{N}\Big(m_j(\mathbf{u}; \mathbf{z}_{\mathrm{pa}(j)}), \tau_j(\mathbf{u})^{-1}\Big), \quad m_j(\mathbf{u}; \mathbf{z}_{\mathrm{pa}(j)}) := \boldsymbol{\lambda}_j(\mathbf{u})^\top \mathbf{z}_{\mathrm{pa}(j)} + \kappa_j(\mathbf{u})/\tau_j(\mathbf{u}),$$

where $\tau_j(\mathbf{u}) = \beta_{\nu,j}^{-1}(\mathbf{u})$ and $\kappa_j(\mathbf{u}) = \tau_j(\mathbf{u})\mu_{\nu,j}(\mathbf{u})$.

Fix any latent realization $(\mathbf{z}_\iota^{(u_0)}, \mathbf{z}_\nu^{(u_0)})$ and draw $\mathbf{u} \sim q_{\mathbf{u}}$ conditioned on $\mathbf{u} \in \mathcal{U}_j$, with Asm. 3.4 ensuring $q_{\mathbf{u}}(\mathcal{U}_j) > 0$. Then one of the following holds:

- *Noise parameters change:* If $(\kappa_j(\mathbf{u}), \tau_j(\mathbf{u})) \neq (\kappa_j(u_0), \tau_j(u_0))$, the two univariate Gaussians for $\mathbf{z}_{\nu,j}$ and $\mathbf{z}_{\nu,j}^{(u_0)}$ (given the same parents) have different mean and/or variance. Since they are continuous and sampled independently in the positive-pair protocol, $\mathbb{P}\big(\mathbf{z}_{\nu,j} = \mathbf{z}_{\nu,j}^{(u_0)} \mid \mathbf{z}_{\mathrm{pa}(j)}\big) = 0$ (by non-degenerate Gaussian), hence $\mathbb{P}\big(\mathbf{z}_{\nu,j} \neq \mathbf{z}_{\nu,j}^{(u_0)}\big) = 1$.

- *Weights change:* If $\boldsymbol{\lambda}_j(\mathbf{u}) \neq \boldsymbol{\lambda}_j(u_0)$, then $m_j(\mathbf{u}; \mathbf{z}_{\mathrm{pa}(j)}) - m_j(u_0; \mathbf{z}_{\mathrm{pa}(j)}) = \big(\boldsymbol{\lambda}_j(\mathbf{u}) - \boldsymbol{\lambda}_j(u_0)\big)^\top \mathbf{z}_{\mathrm{pa}(j)}$. Since $\mathbf{z}_{\mathrm{pa}(j)}$ has a non-degenerate Gaussian distribution, this difference is nonzero with positive probability, making the two conditionals distinct; again, by continuity and independent sampling across the pair, $\mathbb{P}\big(\mathbf{z}_{\nu,j} = \mathbf{z}_{\nu,j}^{(u_0)}\big) = 0$, hence $\mathbb{P}\big(\mathbf{z}_{\nu,j} \neq \mathbf{z}_{\nu,j}^{(u_0)}\big) = 1$.

In both cases, for each $j$ there exist (indeed, with positive probability under $q_{\mathbf{u}}$ there are) environments $\mathbf{u}$ such that

$$\mathbf{z}_\iota = \mathbf{z}_\iota^{(u_0)} \qquad \text{and} \qquad \mathbf{z}_{\nu,j} \neq \mathbf{z}_{\nu,j}^{(u_0)} \quad \text{a.s.} \tag{13}$$

Together with Eq. (12), this implies that for fixed $\mathbf{z}_\iota$ the map $\mathbf{z}_\nu \mapsto h_\iota^\star(\mathbf{z}_\iota, \mathbf{z}_\nu)$ is almost surely constant in the $j$-th coordinate. Since this holds for every $j \in [d_\nu]$, $h_\iota^\star$ is (a.s.) independent of $\mathbf{z}_\nu$, so there exists a measurable $\psi : \mathcal{Z}_\iota \to \mathbb{R}^{d_\iota}$ with

$$h_\iota^\star(\mathbf{z}_\iota, \mathbf{z}_\nu) = \psi(\mathbf{z}_\iota) \qquad \text{a.s.}$$

By the standing regularity at the global maximizer, for any fixed $\mathbf{z}_\nu$ the map $\mathbf{z}_\iota \mapsto f_\iota^\star(\mathbf{z}_\iota, \mathbf{z}_\nu)$ is injective and $C^1$, hence $\psi$ is injective and $C^1$ on $\mathcal{Z}_\iota$. Consequently, there exists a measurable bijection $T : \psi(\mathcal{Z}_\iota) \to \mathbb{R}^{d_\iota}$, and defining $h_\iota := T \circ \psi$ yields

$$f_\iota^\star(\mathbf{x}) = h_\iota(\mathbf{z}_\iota) \qquad \text{a.s.}$$

Therefore $\mathbf{z}_\iota$ is block-identifiable from $f_\iota^\star(\mathbf{x})$ in the sense of Defn. 2.1.

**Step 2 ($\mathbf{z}_\nu$ identifiable with $\mathbf{z}_\iota$ "observed").** From Step 1 we may treat $\mathbf{z}_\iota$ as observed up to a bijection. The responsive block obeys the latent structural equations

$$\mathbf{z}_\nu = \boldsymbol{\lambda}_{\nu\iota}(\mathbf{u})\mathbf{z}_\iota + \boldsymbol{\lambda}_{\nu\nu}(\mathbf{u})\mathbf{z}_\nu + \mathbf{n}_\nu, \qquad \mathbf{n}_\nu \sim \mathcal{N}\big(\boldsymbol{\mu}_\nu(\mathbf{u}), \mathrm{diag}\,\boldsymbol{\beta}_\nu(\mathbf{u})\big), \tag{14}$$

with the anchor $\boldsymbol{\lambda}_{\nu\iota}(u_0) = \mathbf{0}$ and $\boldsymbol{\lambda}_{\nu\nu}(u_0) = \mathbf{0}$ (Asm. 3.1). Hence

$$p(\mathbf{z}_\nu \mid \mathbf{z}_\iota, \mathbf{u}) \propto \exp\Big\{-\tfrac{1}{2}\mathbf{z}_\nu^\top \boldsymbol{\Gamma}(\mathbf{u})\mathbf{z}_\nu + \rho(\mathbf{u}, \mathbf{z}_\iota)^\top \mathbf{z}_\nu\Big\},$$

an exponential family with sufficient statistics $\{\mathbf{z}_\nu, \mathbf{z}_\nu \mathbf{z}_\nu^\top\}$ and natural parameters

$$\boldsymbol{\Gamma}(\mathbf{u}) = \big(\boldsymbol{I} - \boldsymbol{\lambda}_{\nu\nu}(\mathbf{u})\big)^\top \mathrm{diag}\big(\boldsymbol{\tau}(\mathbf{u})\big)\big(\boldsymbol{I} - \boldsymbol{\lambda}_{\nu\nu}(\mathbf{u})\big), \qquad \boldsymbol{\tau}(\mathbf{u}) := \boldsymbol{\beta}_\nu^{-1}(\mathbf{u}),$$

$$\rho(\mathbf{u}, \mathbf{z}_\iota) = \big(\boldsymbol{I} - \boldsymbol{\lambda}_{\nu\nu}(\mathbf{u})\big)^\top \mathrm{diag}\big(\boldsymbol{\tau}(\mathbf{u})\big)\big(\boldsymbol{\mu}_\nu(\mathbf{u}) + \boldsymbol{\lambda}_{\nu\iota}(\mathbf{u})\mathbf{z}_\iota\big).$$

Let $(\phi^\star, f^\star)$ and $(\tilde{\phi}^\star, \tilde{f}^\star)$ be two global maximizers of the joint objective. Because both fit the same $p(\mathbf{x} \mid \mathbf{u})$ and the decoders are diffeomorphisms, their induced conditionals $p(\hat{\mathbf{z}}_\nu \mid \mathbf{z}_\iota, \mathbf{u})$ and $p(\tilde{\mathbf{z}}_\nu \mid \mathbf{z}_\iota, \mathbf{u})$ must coincide with the family above up to a change of variables. By the standard first

step in the proof of Thm. 1 of Liu et al. (2022) (matching the quadratic and linear coefficients across environments), there exist an invertible *constant* matrix $\boldsymbol{A} \in \mathbb{R}^{d_\nu \times d_\nu}$ and vector $\boldsymbol{b} \in \mathbb{R}^{d_\nu}$, both independent of $(\mathbf{z}_\iota, \mathbf{u})$, such that

$$\hat{\mathbf{z}}_\nu = \boldsymbol{A}\,\tilde{\mathbf{z}}_\nu + \boldsymbol{b} \qquad \text{a.s.} \tag{15}$$

*(a) Anchor $\boldsymbol{u}_0$ pins down mixing.* At control $\boldsymbol{u}_0$, Asm. 3.1 gives $\boldsymbol{\lambda}_{\nu\iota} = \boldsymbol{\lambda}_{\nu\nu} = \boldsymbol{0}$, so $\boldsymbol{\Gamma}(\boldsymbol{u}_0) = \mathrm{diag}\big(\boldsymbol{\tau}(\boldsymbol{u}_0)\big)$ is diagonal and $\rho(\boldsymbol{u}_0, \mathbf{z}_\iota) = \mathrm{diag}\big(\boldsymbol{\tau}(\boldsymbol{u}_0)\big)\boldsymbol{\mu}_\nu(\boldsymbol{u}_0)$ is $\mathbf{z}_\iota$-independent. Applying the change of variables $\tilde{\mathbf{z}}_\nu \mapsto \hat{\mathbf{z}}_\nu = \boldsymbol{A}\tilde{\mathbf{z}}_\nu + \boldsymbol{b}$ yields

$$\mathrm{diag}\big(\boldsymbol{\tau}(\boldsymbol{u}_0)\big) = \boldsymbol{A}^\top \widehat{\boldsymbol{\Gamma}}(\boldsymbol{u}_0)\,\boldsymbol{A},$$

with $\widehat{\boldsymbol{\Gamma}}(\boldsymbol{u}_0)$ the (diagonal, positive-definite) precision under the $\tilde{\mathbf{z}}_\nu$-coding. From $\boldsymbol{A}^\top \widehat{\boldsymbol{\Gamma}}(\boldsymbol{u}_0)\boldsymbol{A}$ being diagonal and positive-definite, it follows that $\boldsymbol{A}$ must be a *monomial* matrix, i.e., a *scaled permutation*:

$$\boldsymbol{A} = \boldsymbol{P}\boldsymbol{D}, \qquad \boldsymbol{P} \text{ permutation}, \quad \boldsymbol{D} \succ 0 \text{ diagonal}. \tag{16}$$

*(b) Perturbation richness rules out residual mixing.* By Asm. 3.3, for each node $j \in [d_\nu]$: (i) differences of incoming weights span at each node $j$, which produce independent off-diagonal patterns in $\boldsymbol{\Gamma}(\mathbf{u})$ as $\mathbf{u}$ varies, at least between $\boldsymbol{u}_j$ and $\boldsymbol{u}_0$; and (ii) for each node $j$, there exist $\boldsymbol{u}_j', \boldsymbol{u}_j''$ such that $(\kappa_j(\boldsymbol{u}_j') - \kappa_j(\boldsymbol{u}_0), \tau_j(\boldsymbol{u}_j') - \tau_j(\boldsymbol{u}_0))$ and $(\kappa_j(\boldsymbol{u}_j'') - \kappa_j(\boldsymbol{u}_0), \tau_j(\boldsymbol{u}_j'') - \tau_j(\boldsymbol{u}_0))$ are linearly independent in $\mathbb{R}^2$, giving two independent directions of variation in the diagonal part.

Matching transformed precisions across $\mathbf{u} \in \{\boldsymbol{u}_0, \boldsymbol{u}_j, \boldsymbol{u}_j', \boldsymbol{u}_j''\}$ with Eq. (16) shows that no additional mixing beyond $\boldsymbol{P}\boldsymbol{D}$ is compatible with all constraints; in particular, $\boldsymbol{A}$ cannot depend on $\mathbf{u}$ or $\mathbf{z}_\iota$ and remains $\boldsymbol{P}\boldsymbol{D}$. This mirrors the Step III argument of the proof of Thm 1 in Liu et al. (2022).

*(c) Fixing the shift.* With $\boldsymbol{A} = \boldsymbol{P}\boldsymbol{D}$ fixed, matching the linear terms $\rho(\mathbf{u}, \mathbf{z}_\iota)$ across at least two distinct environments determines a constant shift $\boldsymbol{c}$ such that

$$\hat{\mathbf{z}}_\nu = \boldsymbol{P}\boldsymbol{D}\,\tilde{\mathbf{z}}_\nu + \boldsymbol{c} \qquad \text{a.s.}$$

Taking $\tilde{\mathbf{z}}_\nu \equiv \mathbf{z}_\nu$ yields

$$f_\nu^\star(\mathbf{x}) = \boldsymbol{P}\boldsymbol{D}\,\mathbf{z}_\nu + \boldsymbol{c} \qquad \text{a.s.},$$

which is precisely component-wise identifiability of $\mathbf{z}_\nu$ in the sense of Defn. 2.2.

Therefore, the proof concludes. $\qquad\square$

### B.3 DERIVATION OF THE EVIDENCE LOWER BOUND

In this appendix, we provide a general derivation of the Evidence Lower Bound (ELBO) for our generative model, valid for any intervention vector $\mathbf{u}$.

**Generative model.** For an observation $\mathbf{x}$ under intervention $\mathbf{u}$, the generative model factorizes as:

$$p_\phi(\mathbf{x}, \mathbf{z}_\nu, \mathbf{z}_\iota \mid \mathbf{u}) = p_\phi(\mathbf{x} \mid \mathbf{z}_\nu, \mathbf{z}_\iota)\,p_\phi(\mathbf{z}_\nu \mid \mathbf{u}, \mathbf{z}_\iota)\,p_\phi(\mathbf{z}_\iota), \tag{17}$$

where $\mathbf{z}_\nu$ denotes the *variant* (intervention-specific) latents and $\mathbf{z}_\iota$ the *invariant* latents. The variational posterior adopts the structured mean-field factorization from Eq. 7:

$$q_\theta(\mathbf{z}_\nu, \mathbf{z}_\iota \mid \mathbf{x}, \mathbf{u}) = q_\theta(\mathbf{z}_\nu \mid \mathbf{x}, \mathbf{u})\,q_\theta(\mathbf{z}_\iota \mid \mathbf{x}). \tag{18}$$

**Derivation.** The marginal likelihood is

$$\log p_\phi(\mathbf{x} \mid \mathbf{u}) = \log \int \frac{p_\phi(\mathbf{x}, \mathbf{z}_\nu, \mathbf{z}_\iota \mid \mathbf{u})}{q_\theta(\mathbf{z}_\nu, \mathbf{z}_\iota \mid \mathbf{x}, \mathbf{u})} q_\theta(\mathbf{z}_\nu, \mathbf{z}_\iota \mid \mathbf{x}, \mathbf{u})\,d\mathbf{z}_\nu d\mathbf{z}_\iota.$$

Applying Jensen's inequality to the logarithm yields the ELBO:

$$\begin{aligned}
\mathcal{L}_{\mathrm{ELBO}}(\mathbf{x}, \mathbf{u}) = {}& \mathbb{E}_{q_\theta(\mathbf{z}_\nu, \mathbf{z}_\iota \mid \mathbf{x}, \mathbf{u})}\big[\log p_\phi(\mathbf{x} \mid \mathbf{z}_\nu, \mathbf{z}_\iota)\big] \\
& - \mathbb{E}_{q_\theta(\mathbf{z}_\iota \mid \mathbf{x})}\Big[D_{\mathrm{KL}}(q_\theta(\mathbf{z}_\nu \mid \mathbf{x}, \mathbf{u}) \,\|\, p_\phi(\mathbf{z}_\nu \mid \mathbf{u}, \mathbf{z}_\iota))\Big] \\
& - D_{\mathrm{KL}}(q_\theta(\mathbf{z}_\iota \mid \mathbf{x}) \,\|\, p_\phi(\mathbf{z}_\iota))\,.
\end{aligned} \tag{19}$$

**Modeling interventions.** The intervention vector $\mathbf{u} \in \{0,1\}^M$ is a multi-hot binary vector of dimension $M$, where $M$ is the number of possible targets. - A single-gene perturbation is encoded as a one-hot vector. - A combinatorial perturbation (e.g., genes $j$ and $k$) corresponds to a vector with the $j$-th and $k$-th entries set to 1. - The observational (unperturbed) case is represented by the zero vector $\mathbf{u} = \mathbf{0}$.

Thus, single-gene and multi-gene perturbations are subsumed by the same formulation, and no case-specific ELBO derivations are required.

**Parameterization.** All variational posteriors and priors are chosen as diagonal Gaussians, yielding closed-form KL terms. For example:

$$q_\theta(\mathbf{z}_\iota \mid \mathbf{x}) = \mathcal{N}\big(\boldsymbol{\mu}_\iota(\mathbf{x}), \mathrm{diag}(\boldsymbol{\sigma}_\iota^2(\mathbf{x}))\big),$$

with analogous parameterizations for $q_\theta(\mathbf{z}_\nu \mid \mathbf{x}, \mathbf{u})$, $p_\phi(\mathbf{z}_\nu \mid \mathbf{u})$, and $p_\phi(\mathbf{z}_\iota)$.

### B.4 COEFFICIENT OF DETERMINATION

The coefficient of determination ($R^2$) (Eq. 20) is consistently computed in the observation space, but its interpretation depends on the availability of latent ground truth.

$$R^2 = 1 - \frac{\sum_{i=1}^n (y_i - \hat{y}_i)^2}{\sum_{i=1}^n (y_i - \overline{y})^2} \tag{20}$$

We treat ($R^2 \geq 0.95$ as a successful recovery, indicating alignment within the theoretical identifiability bound.

**Simulation. (Table 1)** In synthetic experiments, we have access to both observed outcomes and the latent variables $\mathbf{z}_\nu, \mathbf{z}_\iota$ that generate them. $R^2$ therefore plays a dual role: it measures predictive accuracy in the observation space and indirectly validates causal recovery, since correctly identified latent factors and structures should yield high predictive performance.

**Real data. (Figure 10)** In experimental single-cell datasets, latent ground truth is unobservable. Here, $R^2$ is computed by comparing the mean expression profiles of generated and real cell populations under the same perturbation condition. Specifically, the model first generates a set of "virtual" cells given a perturbation, from which we compute the mean expression vector across all genes. In parallel, we compute the corresponding mean expression vector from the experimentally observed cells. A linear regression between these two mean vectors yields $R^2$, quantifying how well the generated perturbation response explains the real perturbation response. Thus, in real data, $R^2$ does not directly validate causal recovery but serves as a measure of *practical utility*, assessing whether the learned representations support accurate prediction of population-level transcriptional changes under unseen perturbations.

$R^2$ unifies evaluation across settings: in simulation, it additionally certifies recovery of known latent factors, while in real data it functions as the primary proxy for predictive validity and biological usefulness.

# C ADDITIONAL DETAILS ON EMPIRICAL FINDINGS

## C.1 METHOD DETAILS

We provide details about our training procedure in Algorithm 1

---

**Algorithm 1 Forward and Training Procedure of CDAG-VAE**

---

1: $(\mathbf{x}, \mathbf{u}, \mathbf{x}^{(\boldsymbol{u}_0)}) \sim \mathcal{D}$
2: $\mathbf{h}_1 \leftarrow f_{\mathrm{enc}}(\mathbf{x}); \quad \mathbf{h}_2 \leftarrow f_{\mathrm{enc}}(\mathbf{x}^{(\boldsymbol{u}_0)})$

*— Step 1: Encode Latent Variables —*
3: $(\mu_\nu, \log \sigma_\nu^2) \leftarrow g_\nu(\mathbf{h}_1)$
4: $(\mu_{\iota,1}, \log \sigma_{i,1}^2) \leftarrow g_\iota(\mathbf{h}_1); \quad (\mu_{\iota,2}, \log \sigma_{\iota,2}^2) \leftarrow g_\iota(\mathbf{h}_2)$
5: $\varepsilon_\nu, \varepsilon_{\iota,1}, \varepsilon_{\iota,2} \sim \mathcal{N}(0, I)$
6: $\tilde{\mathbf{z}}_\nu \leftarrow \mu_\nu + \sigma_\nu \odot \varepsilon_\nu; \quad \mathbf{z}_\iota^{(1)} \leftarrow \mu_{\iota,1} + \sigma_{\iota,1} \odot \varepsilon_{\iota,1}$
7: $\mathbf{z}_\iota^{(2)} \leftarrow \mu_{i,2} + \sigma_{\iota,2} \odot \varepsilon_{\iota,2}$

*— Step 2: Structural Equation for $\mathbf{z}_\nu$ —*
8: $\mathbf{W} \leftarrow f_W(\mathbf{u})$ ▷ Adjacency matrix conditioned on soft-intervention
9: $\mathbf{b} \leftarrow B(\mathbf{z}_\iota^{(1)})$ ▷ Contribution from invariant latent
10: $\mathbf{z}_\nu \leftarrow (I - \mathbf{W})^{-1}(\tilde{\mathbf{z}}_\nu + \mathbf{b})$

*— Step 3: Reconstruction —*
11: $\hat{\mathbf{x}} \leftarrow f_{\mathrm{dec}}([\mathbf{z}_\nu, \mathbf{z}_\iota^{(1)}])$

*— Step 4: Loss Calculation —*
12: $\mathcal{L}_{\mathrm{rec}} \leftarrow \|\mathbf{x} - \hat{\mathbf{x}}\|_2^2$
13: $\mathcal{L}_{\mathrm{KL}\text{-}\nu} \leftarrow D_{\mathrm{KL}}(q(\mathbf{z}_\nu|\mathbf{x}) \,\|\, \mathcal{N}(0, I))$
14: $\mathcal{L}_{\mathrm{KL}\text{-}\iota} \leftarrow D_{\mathrm{KL}}(q(\mathbf{z}_\iota^{(1)}|\mathbf{x}) \,\|\, \mathcal{N}(0, I)) + D_{\mathrm{KL}}(q(\mathbf{z}_\iota^{(2)}|\mathbf{x}^{(\boldsymbol{u}_0)}) \,\|\, \mathcal{N}(0, I)$
15: $\mathcal{L}_{\mathrm{contrast}} \leftarrow \mathrm{contrastive}(\mu_{\iota,1}, \mu_{\iota,2})$
16: $(\beta_\nu, \beta_\iota, \alpha) \leftarrow \mathrm{Schedule}(t)$ ▷ Time-dependent annealing schedule
17: $\mathcal{L}_{\mathrm{total}} \leftarrow \mathcal{L}_{\mathrm{rec}} + \beta_\nu \mathcal{L}_{\mathrm{KL}\text{-}\nu} + \beta_\iota \mathcal{L}_{\mathrm{KL}\text{-}\iota} + \alpha \mathcal{L}_{\mathrm{contrast}}$
18: Update $\Theta \leftarrow \Theta - \eta \nabla_\Theta \mathcal{L}_{\mathrm{total}}$

---

## C.2 EXPERIMENT WITH SYNTHETIC DATA

**Basic setup.** We sample data following the DGP described in Sec. 2 with the following details in Table 4.

Table 4: Simulation data generation parameters.

| Quantity | Symbol | Value |
|---|---|---|
| Observation dimension | $\mathbf{x}$ | 500 |
| Latent dimension (variant) | $\mathbf{z}_\nu$ | 4 |
| Latent dimension (invariant) | $\mathbf{z}_\iota$ | 7 |
| Intervention dimension | $\mathbf{u}$ | 12 |
| Training size | – | 3000 |
| Test size | – | 1000 |

**Hyperparameters.** We use the Adam optimizer with hyperparameters detailed in Table 5.

Table 5: Simulation Hyperparameters.

| Hyperparameter | Value | Hyperparameter | Value |
|---|---|---|---|
| Batch size | 64 | $\mathbf{z}_\nu$ dim | 4 |
| Epochs | 100 | $\mathbf{z}_\iota$ dim | 7 |
| Learning rate | $1 \times 10^{-3}$ | $\beta_\nu$ | $1.5 \times 10^{-5}$ |
| $\beta_\iota$ | $5 \times 10^{-4}$ | $\alpha_{\text{contrast}}$ | 0.1 |

**Evaluation metrics.** Identifiability of the variant block $\mathbf{z}_\nu$ is quantified by the mean correlation coefficient (MCC), which measures one-to-one correspondence between each learned latent and its ground-truth counterpart (Def. 2.2). For block-wise disentanglement, we regress the ground-truth latents $(\mathbf{z}_\nu, \mathbf{z}_\iota)$ on their learned estimates $(\hat{\mathbf{z}}_\nu, \hat{\mathbf{z}}_\iota)$ using kernel ridge regression with an RBF kernel, and report the coefficient of determination $(R^2)$. High $R^2$ values close to one indicate block-identifiability (Def. 2.1).

### C.3 STRUCTURE LEARNING

Following Zhang et al. (2023), we first present in Figure 5 the hit map between perturbed genes and the identifiable latent causal components $\mathbf{z}_\nu(i)$ learned by our model. This figure summarizes the dominant associations between external perturbations and latent components: columns correspond to perturbed genes, while rows denote individual causal components. Each entry highlights the component most strongly linked to a given perturbation, thereby revealing how perturbations are distributed across the causal block. This representation facilitates interpretation of the latent space by mapping perturbations onto distinct, identifiable components.

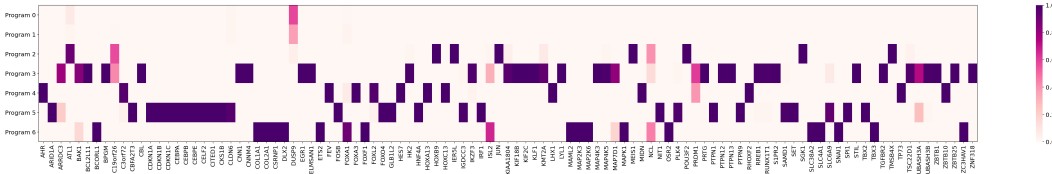

Figure 5: Perturbed gene hits on identifiable causal components.

To further illustrate the structure of the learned causal representation, we visualize the latent causal graph among identifiable components $\mathbf{z}_\nu$. Figure 6 (left) shows the full adjacency matrix estimated by the model (before thresholding), where color intensity reflects the signed effect strength of each edge. For interpretability, we additionally apply a threshold $(\tau = 0.25)$ to prune weak connections, yielding a sparse graph that highlights the dominant causal structure (Figure 6, right). This comparison provides both a complete view of the learned connectivity and a simplified backbone that facilitates biological interpretation.

In Figure 7, we illustrate the inferred causal structure among the latent programs discovered by CDAG-VAE. Each node corresponds to a latent component, and directed edges represent the estimated causal dependencies between them. Importantly, these latent programs can be mapped back to gene-level interpretations, providing biological meaning to the abstract components. For completeness, Table 6 lists the full set of genes associated with each program. This mapping highlights how the learned structure captures both high-level regulatory dependencies and their molecular underpinnings, offering a bridge between statistical causal discovery and biological interpretability.

Beyond the three representative program-level edges discussed in the main text in Figure 4, we provide in Table 7 a summary of the remaining directed edges, together with their mechanistic rationale and supporting references.

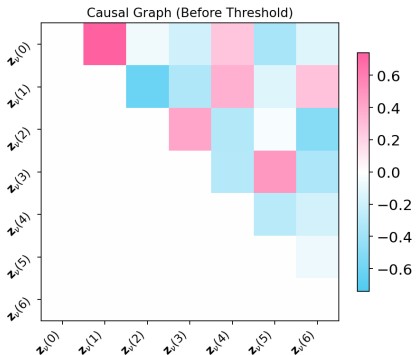 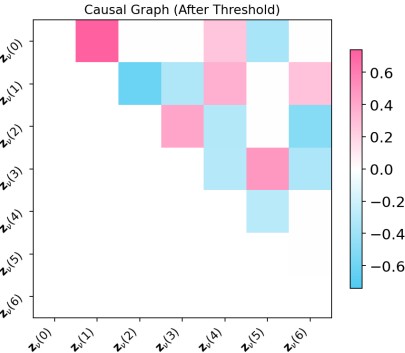

Figure 6: Visualization of the learned causal graph among identifiable components $\mathbf{z}_\nu$. **Left:** full adjacency matrix before thresholding, showing all estimated edges. **Right:** sparse graph after thresholding ($\tau = 0.25$), retaining only dominant edges for interpretability.

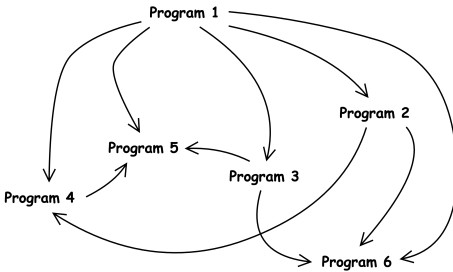

Figure 7: Perturbed gene hits on identifiable causal components.

Table 6: Complete list of genes assigned to each program node inferred from structure learning.

| Program | Genes |
|---------|-------|
| 1 | DUSP9 |
| 2 | ATL1, C19orf26, HOXB9, IER5L, JUN, MEIS1, POU3F2, SGK1, TMSB4X |
| 3 | ARRDC3, BAK1, BCL2L11, BPGM, CBL, CNN1, CNNM4, EGR1, ELMSAN1, HK2, IKZF3, KIAA1804, KIF18B, KIF2C, KLF1, KMT2A, LYL1, MAP4K3, MAP4K5, MAP7D1, PRDM1, PRTG, PTPN12, PTPN13, RREB1, RUNX1T1, S1PR2, STIL, TGFBR2, TSC22D1, UBASH3A, UBASH3B, ZBTB1, ZBTB25, ZNF318 |
| 4 | AHR, C3orf72, FEV, FOXA3, FOXL2, HES7, HOXA13, HOXC13, LHX1, MIDN, RHOXF2, TP73, ZBTB10 |
| 5 | ARID1A, CBFA2T3, CDKN1A, CDKN1B, CDKN1C, CEBPA, CEBPB, CEBPE, CELF2, CITED1, CKS1B, CLDN6, FOSB, FOXO4, GLB1L2, HNF4A, IGDCC3, IRF1, NIT1, PLK4, PTPN1, PTPN9, SAMD1, SET, SLC6A9, SPI1, TBX2 |
| 6 | BCORL1, COL1A1, COL2A1, CSRNP1, DLX2, ETS2, FOXA1, FOXF1, ISL2, MAML2, MAP2K3, MAP2K6, MAPK1, NCL, OSR2, SLC38A2, SLC4A1, SNAI1, TBX3, ZC3HAV1 |

Table 7: Program-level representative edges: mechanistic rationale and supporting references.

| Edge | Mechanistic rationale (summary) | Refs. |
|---|---|---|
| DUSP9 → TGFBR2 | TGFBR2 activates ERK through a non-Smad branch; DUSP9 dephosphorylates ERK/JNK, attenuating this output. | (Emanuelli et al., 2008) (Zhang, 2009) |
| DUSP9 → TP73 | c-Jun enhances TP73 stability and activity; DUSP9 lowers JNK/ERK→AP-1 signaling, indirectly downregulating TP73. | (Koeppel et al., 2011) (Emanuelli et al., 2008) |
| DUSP9 → CDKN1A | ERK → ELK1/EGR1 induces p21 transcription; DUSP9 suppresses ERK phosphorylation, blunting this induction. | (Lim et al., 1998) (Ragione et al., 2003) |
| DUSP9 → SNAI1 | Epithelial–mesenchymal transition (EMT) induction requires SMAD3–AP-1 cooperation; DUSP9 attenuates AP-1, weakening SNAI1 transcription. | (Sundqvist et al., 2013) (Fan et al., 2025) |
| JUN → TP73 | c-Jun stabilizes and potentiates TP73, enhancing apoptosis-related transcription. | (Koeppel et al., 2011) |
| JUN → SNAI1 | AP-1 (c-Jun) cooperates with SMAD factors to elevate SNAI1 expression in TGF-$\beta$-driven EMT. | (Sundqvist et al., 2013) (Fan et al., 2025) |
| TGFBR2 → CDKN1A | Canonical SMAD2/3/4 downstream of TGFBR2 transactivates p21, enforcing cytostasis. | (Ikushima & Miyazono, 2010) |

## C.4 UNPERTURBED LATENT SPACE

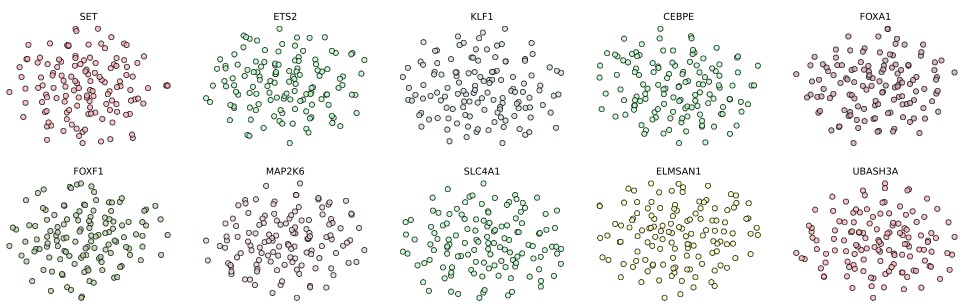

Figure 8: t-SNE visualization of invariant block $\mathbf{z}_\iota$ for 10 single-gene perturbations.

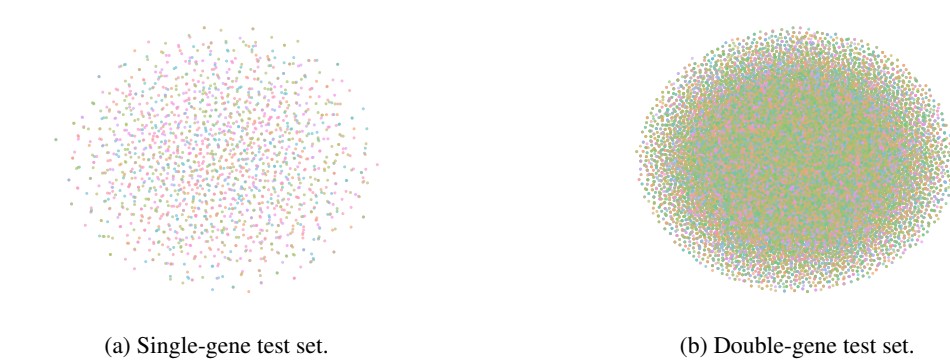

(a) Single-gene test set.         (b) Double-gene test set.

Figure 9: t-SNE visualization of the invariant block $z_\iota$ for single-gene (a) and double-gene (b) perturbation conditions in the test set.

We further report additional t-SNE projections of the invariant block $\mathbf{z}_\iota$. Fig. 8 presents the latent spaces for all remaining single-gene perturbations in the test set, complementing the representative examples shown in the main text. Figure 9 further shows the t-SNE embeddings for the entire single-gene and double-gene test sets. Across all settings, cells from distinct perturbation conditions remain well-mixed rather than forming separate clusters, providing additional evidence that $\mathbf{z}_\iota$ captures perturbation-invariant background transcriptional programs.

### C.5 EXTENDED EXPERIMENTS AND ADDITIONAL RESULTS ON REAL DATA (ALL GENES)

**Hyperparameter settings for real data experiments.** We use the Adam optimizer with hyperparameters detailed in Table 8.

Table 8: Real Data Hyperparameter.

| Hyperparameter | Value |
|---|---|
| Batch size | 64 |
| Epochs | 100 |
| Learning rate | $1 \times 10^{-4}$ |
| Hidden dimension | 256 |
| $\mathbf{z}$ dimension | 10, 35, 75, 100 |
| $\alpha_{\text{contrast}}$ | 0.05 |
| $\beta_\nu, \beta_\iota$ | $1 \times 10^{-2}$ |

**Results on Single-Gene Perturbation Prediction.** Table 9 reports the RMSE and Figure B.4 illustrates the $R^2$ performance of CDAG-VAE on single-gene perturbation prediction across different latent dimensionalities. We experimented with four latent configurations: $(z_\nu, z_\iota) \in \{(4, 6), (7, 28), (15, 60), (20, 85)\}$, corresponding to total latent dimensionalities $z \in \{10, 35, 75, 100\}$. These settings enforce $z_\nu < z_\iota$, reflecting the modeling assumption that perturbation-responsive variation resides in a lower-dimensional subspace compared to invariant background programs.

Across all settings, CDAG-VAE consistently achieved the best performance relative to baselines. On RMSE, our model yielded the lowest reconstruction error, highlighting its fidelity in capturing single-gene expression responses. On $R^2$, CDAG-VAE attained values close to 1.0, demonstrating robust predictive accuracy. Performance remained stable as dimensionality increased, indicating that the framework is not overly sensitive to the precise choice of $z_\nu$ and $z_\iota$, as long as the variant subspace is smaller than the invariant one. Together, these results validate that explicitly disentangling perturbation-responsive and invariant subspaces yields clear empirical advantages for single-gene perturbation prediction.

**CDAGVAE MMD variant.** To complement the main experiments, we evaluate a maximum mean discrepancy (MMD)-based variant of our model, denoted as CDAG-VAE(MMD). This variant augments the objective with an MMD regularization term to enforce distributional alignment, similar to the approach in Zhang et al. (2023). This allows us to fairly compare the proposed model with the existing Discrepancy-VAE from Zhang et al. (2023) using MMD-based metrics. For completeness, we report its performance on single-gene perturbation benchmarks in Table 10.

Table 9: RMSE on single-gene perturbation prediction.

| Method | Latent dimension | | | |
|---|---|---|---|---|
| | **10** | **35** | **75** | **105** |
| **Discrepancy-VAE** (Zhang et al., 2023) | $0.5603_{\pm 0.0030}$ | $0.5560_{\pm 0.0027}$ | $0.5582_{\pm 0.0038}$ | $0.5558_{\pm 0.0022}$ |
| **SENA** (de la Fuente et al., 2025) | $0.5839_{\pm 0.0021}$ | $0.5837_{\pm 0.0086}$ | $0.5778_{\pm 0.0109}$ | $0.5837_{\pm 0.0074}$ |
| **sVAE+** (Lopez et al., 2022) | $0.5012_{\pm 0.0018}$ | $0.5005_{\pm 0.0025}$ | $0.5003_{\pm 0.0024}$ | $0.5002_{\pm 0.0022}$ |
| **SAMS-VAE** (Bereket & Karaletsos, 2023) | $0.4114_{\pm 0.0020}$ | $0.4136_{\pm 0.0019}$ | $0.4140_{\pm 0.0022}$ | $0.4123_{\pm 0.0290}$ |
| **DAG-VAE (Ours)** | $0.4098_{\pm 0.0001}$ | $0.4115_{\pm 0.0008}$ | $0.4115_{\pm 0.0005}$ | $0.4155_{\pm 0.0038}$ |
| **cDAG-VAE (Ours)** | $0.4027_{\pm 0.0028}$ | $0.3998_{\pm 0.0013}$ | $0.3997_{\pm 0.0013}$ | $0.3995_{\pm 0.0013}$ |

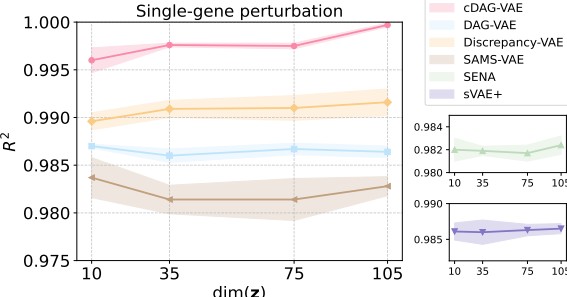

Figure 10: $R^2$ on single-gene perturbation

Table 10: Evaluation of the cDAG-VAE with MMD variant on single-gene perturbation prediction.

| Method | Metrics | | |
|---|---|---|---|
| | **RMSE** | **$R^2$** | **MMD** |
| **Discrepancy-VAE** (Zhang et al., 2023) | $0.5558_{\pm 0.0022}$ | $0.9916_{\pm 0.0014}$ | $0.3243_{\pm 0.0050}$ |
| **cDAG-VAE** (MMD) | $0.5485_{\pm 0.0013}$ | $0.9958_{\pm 0.0003}$ | $0.3077_{\pm 0.0036}$ |

**Ablation on Latent Capacity Allocation.** Our ablation studies show that asymmetric allocation of latent capacity is crucial, with the invariant block ($\mathbf{z}_\iota$) serving as the primary bottleneck. As reported in Table 2 and Table 9, together with Figure 10 and Figure 3, the invariant-heavy configuration $((20, 85)$; total $\mathbf{z} = 105)$ clearly outperforms alternative splits, achieving the lowest RMSE and highest $R^2$ on both in-distribution and out-of-distribution predictions. This suggests that sufficient capacity for modeling background transcriptional states is critical.

In contrast, when $\mathbf{z}_\iota$ is under-resourced—such as in the variant-heavy setting ($z_\nu = 85, z_\iota = 20$) or the equal-split setting ($\mathbf{z}_\nu = 50, \mathbf{z}_\iota = 55$)—performance declines noticeably, with outcomes that are largely indistinguishable (Table 11). These results suggest two observations: (1) in our tested configurations, the variant block $\mathbf{z}_\nu$ already appears adequate at relatively small dimensionalities, and allocating further capacity beyond this does not yield additional gains; and (2) the invariant block $\mathbf{z}_\iota$ is the performance-limiting factor, as reduced capacity creates a bottleneck that additional $\mathbf{z}_\nu$ dimensions are insufficient to compensate for.

Table 11: Results on single- and double-gene perturbations under different capacity allocations of $\mathbf{z}_\nu$ and $\mathbf{z}_\iota$

| Dimension | Single-Gene Perturbation | | Double-Gene Perturbation | |
|---|---|---|---|---|
| | RMSE | $R^2$ | RMSE | $R^2$ |
| $\mathbf{z}_\nu = \mathbf{z}_\iota$ | $0.4084_{\pm 0.0011}$ | $0.9875_{\pm 0.0007}$ | $0.4627_{\pm 0.0003}$ | $0.9649_{\pm 0.0003}$ |
| $\mathbf{z}_\nu > \mathbf{z}_\iota$ | $0.4084_{\pm 0.0010}$ | $0.9875_{\pm 0.0007}$ | $0.4627_{\pm 0.0002}$ | $0.9649_{\pm 0.0002}$ |
| $\mathbf{z}_\nu < \mathbf{z}_\iota$ | $0.3995_{\pm 0.0013}$ | $0.9977_{\pm 0.0002}$ | $0.4474_{\pm 0.0007}$ | $0.9865_{\pm 0.0009}$ |

Together, these findings align with biological intuition: accurately representing cellular identity requires a high-capacity invariant subspace $\mathbf{z}_\iota$, reflecting the complexity of background transcriptional programs, whereas a comparatively smaller variant subspace $\mathbf{z}_\nu$ suffices to capture the sparse, perturbation-specific effects.

**Ablation on Contrastive Alignment.** We further ablated the alignment term by comparing CDAG-VAE with and without the alignment loss ($\alpha = 0.05$ vs. $\alpha = 0$) under a fixed latent dimension ($\mathbf{z} = 105$). Results (Figure 12) consistently show that including the alignment term improves performance across both single- and double-gene perturbation prediction.

In particular, when $\alpha = 0$, the invariant block $\mathbf{z}_\iota$ collapses, carrying little information (empirically $\mathrm{KL}_i \to 0$), and the effective latent capacity is dominated by the variant block $\mathbf{z}_\nu$. As a result, performance under $\alpha = 0$ closely resembles that of capacity splits with $\mathbf{z}_\nu \geq \mathbf{z}_\iota$, where the model effectively ignores the invariant subspace. In contrast, with $\alpha = 0.05$, the alignment signal enforces informativeness of $\mathbf{z}_\iota$, preventing leakage of perturbation-specific effects into the invariant block. This leads to consistently better generalization, especially on out-of-distribution double-gene conditions.

Our results indicate that the contrastive alignment loss is important for sustaining the informativeness of the invariant block and maintaining block disentanglement. Even under fixed total latent capacity, models with the alignment loss consistently achieve higher accuracy, suggesting that alignment is a key component for reliable generalization in CDAG-VAE.

Table 12: Single- and double-gene performance under contrastive alignment ablation.

| Contrastive Alignment | Single-Gene Perturbation | | Double-Gene Perturbation | |
|---|---|---|---|---|
| | RMSE | $R^2$ | RMSE | $R^2$ |
| ✗ | $0.4083_{\pm 0.0011}$ | $0.9875_{\pm 0.0007}$ | $0.4626_{\pm 0.0002}$ | $0.9650_{\pm 0.0002}$ |
| ✓ | $0.3995_{\pm 0.0013}$ | $0.9977_{\pm 0.0002}$ | $0.4474_{\pm 0.0007}$ | $0.9865_{\pm 0.0009}$ |

## C.6 VALIDATING CONTRASTIVE DISENTANGLEMENT ON DIFFERENTIALLY EXPRESSED GENES

**Metric Definitions and Empirical Observations** To more finely assess the model's fidelity in capturing biologically meaningful perturbation effects beyond aggregate statistics, we compute performance metrics on two complementary feature sets for each perturbation condition:

- **All genes**: measurements computed using the entire 5,000-dimensional gene expression vectors, reflecting the global cellular state.

- **DE genes**: measurements computed using the 20-dimensional sub-vectors corresponding to the top 20 most differentially expressed genes.

We make the following empirical observations:

- **In-distribution (single-gene).** The model achieves high accuracy on both feature sets. The $R^2$ scores for "DE genes" are nearly identical to the global "All genes" $R^2$, while the RMSE on the DE subset is notably lower than the global average (Figure 12).

- **Out-of-distribution (double-gene).** While the global "All genes" $R^2$ remains consistently high (around $\sim 0.98$), the $R^2$ on the "DE genes" subset exhibits a mild degradation, with a fraction of perturbations showing scores in the 0.5–0.9 range. The DE-gene RMSE is typically lower than or comparable to the "All genes" RMSE, though a subset of double-gene conditions exhibits higher deviation in the DE subspace, reflecting the increased complexity of specific combinatorial interactions (Figure 11).

**Interpretation via contrastive disentanglement.** These patterns are broadly consistent with the intended disentanglement mechanism of cDAG-VAE.

**Successful modeling of invariant background ($\mathbf{z}_\iota$).** The persistently high $R^2$ on the 5,000-dimensional "All genes" vectors suggests that the contrastive alignment term effectively stabilizes *background cellular programs* across perturbations. Since the vast majority of genes exhibit relatively small perturbation effects and are primarily governed by such background programs, the model's ability to reconstruct the global transcriptomic state—in both single- and double-gene settings—indicates that the invariant latent factors $\mathbf{z}_\iota$ capture a robust, perturbation-stable representation rather than overfitting to individual conditions.

**Causal uncertainty concentrated in the perturbation-responsive subspace ($\mathbf{z}_\nu$).** By construction, our model is designed so that perturbation-responsive variation is represented in the variant latent block $Z_\nu$, while "DE genes" form a small, perturbation-enriched readout of this subspace. The fact that $R^2$ on DE genes degrades more noticeably than $R^2_{\text{All}}$ under double-gene OOD prediction reflects the inherent difficulty of zero-shot combinatorial causal extrapolation, where novel, potentially non-additive interactions must be inferred from single-gene training data. At the same time, the observation that DE-gene RMSE typically remains low—despite reduced $R^2_{\text{DE}}$ for a subset of double perturbations—suggests that the model often predicts the *magnitude* of key expression changes reasonably well, even when finer-grained variance patterns are harder to match in a zero-shot setting.

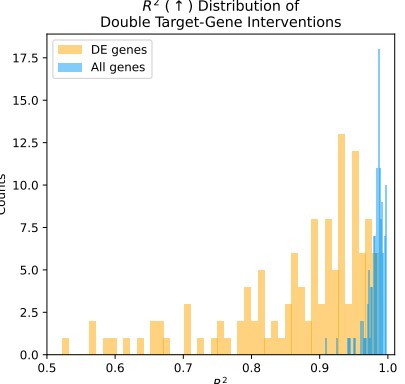
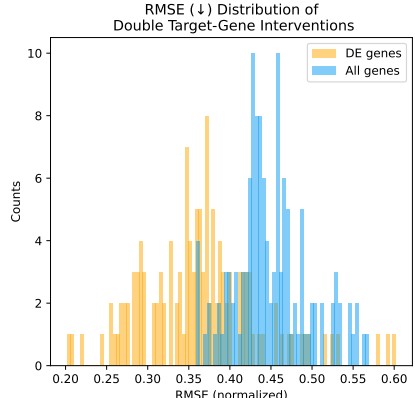

Figure 11: Results of double-gene perturbation on DE genes.

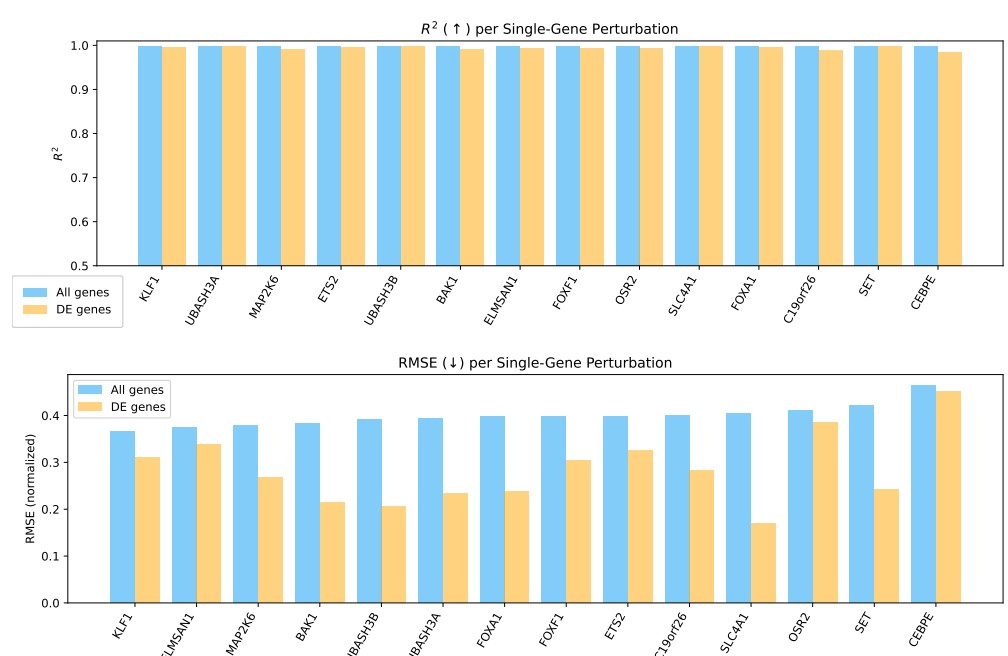

Figure 12: Results of single-gene perturbation on DE genes.

## C.7 PERSPECTIVE ON LATENT CAUSAL MODEL FOR DOUBLE-GENE PERTURBATION

Recent benchmarking results (Ahlmann-Eltze et al., 2025) have brought renewed clarity to the structural characteristics of perturbation–effect prediction. On the Norman2019 dataset, the authors showed that even sophisticated architectures—including GEARS (Roohani et al., 2024) and several foundation-model variants—often fail to outperform a simple additive baseline when evaluated on pseudobulk expression responses to double perturbations. This outcome reflects an important property of the benchmark: for high-expression genes, the dominant component of the double-perturbation signal is well approximated by a linear superposition of single-gene log-fold changes, leaving limited opportunity for complex representation-heavy models to demonstrate gains under squared-error metrics.

Our work, however, differs fundamentally from this regression-centric setting. Rather than optimizing directly for pseudobulk reconstruction, we aim to learn *latent causal factors* that enable mechanism-level disentanglement and robust generalization to combinatorial interventions without any supervision on double perturbations. Nonetheless, the benchmark raises two questions that are highly pertinent to Causal Representation Learning (CRL): (i) under a strict OOD protocol in which *no* double-perturbation data are available during training, do classical linear baselines retain their apparent advantage? (ii) beyond explaining variance in high-expression pseudobulk profiles, can a structured latent model more faithfully recover the Average Treatment Effect (ATE) at the perturbation-label level, thereby distinguishing deterministic causal responses from stochastic single-cell noise?

A central distinction between our CRL approach and regression-based predictors lies in the underlying data-generating process (DGP) being modeled. Rather than mapping perturbations directly to high-dimensional gene expression vectors, our model assumes that observations arise from a set of low-dimensional latent causal variable $\mathbf{z}$ whose dynamics are modulated by interventions $u$ and corrupted by biologically meaningful stochasticity $\mathbf{n}$. In this formulation, $\mathbf{z}$ does not represent gene expression itself, but instead captures cellular programs, pathway activities, or regulatory modules that mediate the effect of perturbations. The observed expression $\mathbf{x}$ is treated as a nonlinear projection of these latent factors through the decoding mechanism of the VAE.

The noise term $n$ plays an equally important conceptual role. It reflects the substantial cell-to-cell stochasticity inherent in single-cell transcriptomics, including transcriptional bursting, technical

variation, and biologically unstructured fluctuations not explained by the regulatory graph. By explicitly modeling this DGP rather than collapsing the data into pseudobulk averages, our method aims to separate deterministic causal responses from stochastic variation, enabling latent mechanisms to be identifiable and supporting robust generalization to unseen combinatorial perturbations.

**Feature Space** We evaluate model performance on two complementary gene sets to balance standard comparability with causal validity.

**High-Expression Benchmark Subset.** Following the protocol of Ahlmann-Eltze et al. (2025), we first compute metrics on the 1,000 most highly expressed genes in control cells. This subset represents a stable, high–signal-to-noise regime and serves as the standard benchmark for pseudobulk perturbation prediction, specifically for comparing deep learning methods against linear baselines like the additive model.

**Genome-wide Expression Profile.** To validate the model's capacity to capture the full regulatory landscape, we focous on evaluating performance on the Genome-wide Expression Profile. This assessment aligns directly with the core design objective of our cDAGVAE: to identify and disentangle the latent background cellular programs that underpin biological processes. Crucially, these programs often manifest as pervasive but subtle signals—residing in low-abundance regimes or buried within technical noise, that are systematically excluded by top-expression filters. Restricting evaluation to high-expression genes would therefore risk measuring only the dominant perturbation effects while overlooking these intricate background dynamics. Genome-wide evaluation is thus essential to verify that the model has successfully recovered these weak yet fundamental cellular programs across the full dynamic range of the transcriptome.

**Evaluation Granularity** To provide a rigorous and biologically grounded assessment, we report performance at two complementary levels of granularity: condition-level pseudobulk averages and cell-level Heterogeneity.

**Condition-level Pseudobulk Averages.** Following the benchmarking protocol of Ahlmann-Eltze et al. (2025), we aggregate single-cell expression profiles within each perturbation into a pseudobulk vector by averaging across cells. Metrics computed on these condition-level profiles (e.g.Delta Pearson, $L2$, RMSE, $R^2$) quantify how well a model recovers the average transcriptional response associated with each perturbation. This aggregation suppresses stochastic technical noise and cell-to-cell variability, yielding a high–signal-to-noise summary that captures the dominant regulatory signature. As such, pseudobulk-based evaluation serves as the standard reference for regression-style perturbation–effect prediction and provides a direct point of comparison to linear baselines such as the additive model.

**Cell-level Heterogeneity Evaluation.** Unlike standard pseudobulk metrics, which deliberately average away cell-to-cell heterogeneity, our evaluation is designed to probe how well a model explains the distribution of single-cell states under each perturbation. For every perturbation label $u$, the model produces a predicted mean expression vector, which we treat as a deterministic summary of $p_\theta(\mathbf{x} \mid \mathbf{u} = u)$. We then compare this predicted mean against the full ensemble of observed single-cell profiles assigned to $u$, computing RMSE and $R^2$ at the single-cell level with respect to the condition-mean baseline, and finally averaging these scores across held-out double-perturbation conditions. In contrast to purely pseudobulk-based metrics, this *perturbation-conditioned single-cell evaluation* directly measures how well the model reconciles biological noise with the structured heterogeneity induced by different interventions.

This perspective is especially important for our contrastive latent causal generative model, whose primary goal is to decompose perturbation-driven heterogeneity rather than merely reproduce bulk-like signatures. In cDAG-VAE, the invariant block $\mathbf{z}_\iota$ is trained to capture shared background cellular programs that persist across perturbations, while the variant block $\mathbf{z}_\nu$ encodes perturbation-responsive mechanisms that shift the distribution of single-cell states in a condition-specific manner. Strong performance under the perturbation-conditioned single-cell metric therefore indicates that the learned latent space has disentangled these two sources of variability: $\mathbf{z}_\iota$ provides a stable scaffold for global cellular state, and $\mathbf{z}_\nu$ systematically explains how different perturbations reshape the high-dimensional expression landscape, particularly in the DE-gene–enriched subspaces analyzed in App. C.6. From a single-cell bioinformatics standpoint, this means that cDAG-VAE does not merely fit average responses, but learns a coherent generative model of across-perturbation single-

cell heterogeneity, supporting downstream tasks such as mechanistic interpretation and zero-shot generalization to unseen combinatorial perturbations.

Table 13: Supplementary robustness evaluation on Genome-wide expression profile.

| Method | Condition-level | | | | Cell-level | |
|---|---|---|---|---|---|---|
| | Prediction error ($L2$) | Pearson Delta | RMSE | $R^2$ | RMSE | $R^2$ |
| Additive | $2.5407_{\pm 0.0000}$ | $0.9076_{\pm 0.0000}$ | $0.0887_{\pm 0.0000}$ | $0.6431_{\pm 0.0000}$ | $0.4424_{\pm 0.0000}$ | $-$ |
| GEARS | $4.6797_{\pm 0.2620}$ | $0.4631_{\pm 0.0644}$ | $0.1514_{\pm 0.0086}$ | $0.9730_{\pm 0.0032}$ | $0.5861_{\pm 0.0031}$ | $-$ |
| cDAGVAE | $3.7238_{\pm 0.0012}$ | $0.6869_{\pm 0.0005}$ | $0.1285_{\pm 0.0015}$ | $0.9965_{\pm 0.0005}$ | $0.4494_{\pm 0.0008}$ | $0.9840_{\pm 0.0011}$ |

*Note.* A dash (–) indicates that the model yields a negative $R^2$, it performs worse than a trivial

mean predictor. Exact magnitudes are omitted because they have no interpretable biological meaning in this setting.

Table 14: Supplementary robustness evaluation on High-expression Genes.

| Method | Condition-level | | | | Cell-level | |
|---|---|---|---|---|---|---|
| | Prediction error ($L2$) | Pearson Delta | RMSE | $R^2$ | RMSE | $R^2$ |
| Additive | $2.4906_{\pm 0.0000}$ | $0.9101_{\pm 0.0000}$ | $0.0870_{\pm 0.0000}$ | $0.6470_{\pm 0.0000}$ | $0.4332_{\pm 0.0000}$ | $-$ |
| GEARS | $4.2649_{\pm 0.2044}$ | $0.5068_{\pm 0.0710}$ | $0.1381_{\pm 0.0065}$ | $0.9682_{\pm 0.0065}$ | $0.5746_{\pm 0.0018}$ | $-$ |
| cDAGVAE | $3.6491_{\pm 0.0010}$ | $0.6936_{\pm 0.0004}$ | $0.1259_{\pm 0.0013}$ | $0.9951_{\pm 0.0005}$ | $0.4411_{\pm 0.0007}$ | $0.9758_{\pm 0.0011}$ |

Tables 13–14 report the performance of cDAG-VAE, the additive baseline, and GEARS on both the genome-wide expression profiles and the high-expression gene subset. Focusing on the deep learning models, cDAG-VAE achieves higher $R^2$ and lower RMSE than GEARS under our single-gene $\rightarrow$ double-gene OOD evaluation, both for the full transcriptome and for the high-expression subset. Within this strictly single-to-double OOD setting, these gains indicate that conditioning prediction on a learned causal latent representation of single-gene perturbations can more effectively support generalization to unseen double perturbations than directly learning a perturbation-to-expression mapping with the graph neural network baseline GEARS.

In line with the report of Ahlmann-Eltze et al. (2025), the simple additive baseline remains highly competitive on condition-level pseudobulk metrics. In our experiments, it achieves the lowest $L_2$ error and the highest Delta Pearson on pseudobulk profiles, especially on the High-Expression Gene subset on which the benchmark was originally defined. This behavior is unsurprising on Norman2019: for many gene pairs, the dominant component of the condition-level response is well approximated by a linear superposition of single-gene effects, which matches the inductive bias built into the additive model. By contrast, deep models such as GEARS and cDAG-VAE must recover this approximate linearity from data while also representing residual non-linear interactions and higher-order structure. Under purely average-effect metrics such as pseudobulk $L_2$, this additional flexibility can manifest as a small performance gap relative to the hard-coded additive baseline, even when the deep models offer clear advantages at the single-cell and out-of-distribution evaluation levels.

However, relying solely on condition-level error obscures an important distinction between linear baselines and causal generative models. Despite its strong $L_2$ and Delta Pearson performance, the additive model attains negative cell-level $R^2$ on Norman2019, similar to GEARS; on average, both methods offer little or no improvement over predicting each cell by its condition mean when evaluated against the full single-cell population. In contrast, cDAG-VAE achieves substantially higher cell-wise $R^2$ (often close to 1.0), indicating that it explains a large fraction of cell-specific variance across cells while remaining highly competitive at the pseudobulk level. This pattern reflects a difference in modeling objectives: in our setup, the additive model and GEARS are trained and evaluated primarily as regression estimators of the average conditional response $\mathbb{E}[\mathbf{x} \mid \mathbf{u}]$, whereas cDAGVAE is a generative causal model that explicitly targets the underlying conditional distribution $p(\mathbf{x} \mid \mathbf{u})$ of single-cell expression given the perturbation condition $\mathbf{u}$. By learning disentangled latent factors that encode both background cellular programs and perturbation-responsive mechanisms, cDAG-VAE can match linear baselines on condition-level metrics while more accurately capturing how double perturbations reshape the single-cell state distribution. For CRL, such single-cell–level fidelity is

crucial for downstream tasks including mechanism interpretation, causal structure discovery, and robust OOD generalization.

# D  LARGE LANGUAGE MODEL USAGE

We disclose the use of large language models (LLMs) in the preparation of this manuscript. Their use was strictly limited to improving the clarity and style of the language, as well as assisting in formulating search queries for literature review. All core scientific contributions are exclusively human-generated, including the formulation of the research problem, the design of the methodology, theoretical proofs, experimental implementation, and analysis of results. LLMs were not used to generate scientific content such as methods, results, or arguments. All cited works were independently sourced, read, and verified by the authors. The authors carefully reviewed all LLM-assisted text and bear full responsibility for the accuracy and integrity of the manuscript. No confidential or unpublished data were shared with any LLM service.

