# OpenReview forum: "Identifying Unperturbed Cellular Programs Enables Accurate Single-Cell Perturbation Prediction"
_ICLR.cc/2026/Conference — Submitted to ICLR 2026_

### Official Review · Reviewer_bvpL · 2025-10-15

**Soundness:** 2
**Presentation:** 3
**Contribution:** 2
**Rating:** 2
**Confidence:** 3

**Summary:**

This paper proposes a novel generative model (cDAG-VAE) for predicting the effects of gene perturbations in single-cell transcriptomic data. The model explicitly disentangles the latent space into variant factors that capture perturbation-induced changes and invariant factors that capture background cellular programs. The authors provide theoretical identifiability results via a differentiability analysis and introduce a contrastive alignment loss to enforce the causal structure.

**Strengths:**

The idea of disentangling the latent representation into perturbation-responsive and invariant subspaces is addressing a known & important challenge in perturbation modeling, previously tackled by SAMS-VAE (Bereket & Karaletsos, 2023), discrepancy-VAE (Zhang et al., 2023) and sVAE (Lopez et al., 2022).

**Weaknesses:**

1. The empirical comparison is restricted to other VAE-based methods and omits simple statistical and non-VAE baselines. Additionally, the paper does not include a comparison against SAMS-VAE (“Modeling Cellular Perturbations with the Sparse Additive Mechanism Shift Variational Autoencoder”, Bereket et al., 2023), which was specifically designed for interpretable perturbation modeling with causal structure. No linear or GLM-based approaches are tested. Recent work by Huber et al. in “Evaluating Deep Models for Predicting Single-Cell Perturbation Outcomes” (Nature Methods, 2025) highlights that deep learning models often fail to outperform trivial linear baselines in perturbation prediction. The absence of even a basic additive or no-change baseline, as used in “Systema: A Comprehensive Benchmark for Single-Cell Perturbation Modeling” (Roohani et al., Nature Biotechnology, 2024) and “Benchmarking Foundation Models for Cell Perturbation Prediction” (Szalai et al., BMC Genomics, 2025), shows a serious gap. Similarly, specialized models such as GPerturb from “Gaussian Process Modeling of Single-Cell Perturbation Responses” (Koh et al., 2023), or alternative ML architectures (e.g., transformers or graph models), are entirely ignored. Transformer-based models like scGPT and scBERT, and graph-based models like GEARS (introduced in “Predicting Transcriptional Outcomes of Novel Multigene Perturbations”, Roohani et al., 2024), were shown to match or outperform VAE-based methods in Systema. Without comparisons to these broader approaches, it remains unclear whether the reported improvements stem from model novelty or simply from benchmark favorability.

2. The paper relies exclusively on RMSE and R² for performance evaluation, both of which are problematic when applied to high-dimensional gene expression space. These metrics are often dominated by expression of highly variable or abundant genes and may obscure biologically meaningful differences. In contrast, recent benchmarks employ more informative metrics for causal accuracy, such as Pearson correlation on top differentially expressed genes, or precision in identifying the perturbed target gene itself. The current paper does not include any such metrics, e.g., accuracy of identifying the perturbed gene, which are essential to support claims about “accurate perturbation prediction” or causal interpretability.

3. All experiments are performed on the Norman2019 dataset, a CRISPRa Perturb-seq screen in K562 cells with 105 single-gene and 131 combinatorial perturbations. While this is a valuable dataset, its exclusive use is problematic. No results are presented on other datasets such as the CRISPRi screens.

4. The paper repeatedly claims to recover the “true causal impact” of perturbations and implies that the learned variant latent factors reflect genuine causal mechanisms. If the model’s latent directed acyclic graph (DAG) is intended to reflect biological causal structure, then it should be evaluated using standard gene regulatory network (GRN) inference benchmarks. Despite generating a DAG over latent variables, the paper provides no quantitative comparison to existing GRN inference methods or ground truth networks (e.g., from ChIP-seq, transcription factor databases, or curated pathway maps). Without such validation, it's unclear whether the learned graph reflects meaningful biological regulation or simply captures statistical variation in the latent space.

5. The identifiability results in Section 3 rely on strong and biologically questionable assumptions such as a smooth, invertible generative map and perturbation richness across all latent dimensions. In reality, gene expression is noisy, sparse, and often affects only a subset of factors. Moreover, the theoretical guarantees assume global optimization of a non-convex objective, with no discussion of convergence or robustness. These assumptions are not tested in practice and should be more clearly acknowledged.

**Questions:**

1. Why are no statistical baselines included in the comparison?
2. Can the authors evaluate on gene regulatory network inference benchmarks if the goal is learning a DAG? And how can we interpret the learned DAG without validation against known regulatory networks?
3. How often does the model correctly identify the perturbed gene or DE target?
4. Why are biologically meaningful metrics such as Pearson correlation on top differentially expressed genes not reported?
5. Why is SAMS-VAE (Bereket et al., 2023) excluded from the benchmark despite its relevance?

**Details Of Ethics Concerns:**

No ethics concerns.

---

> ### Author Response · Authors · 2025-11-19
>
> `W&Q 1 `
>
> ***We tried to look up the suggested references but were unable to locate most of them in public repositories.***
>
> Our work primarily targets **causal representation learning (CRL)** and  **identifiability**, rather than solely improving predictive accuracy. Accordingly, our baseline design follows this objective.
>
>
> • ***Top-level rationale for baseline selection***
>
> Our baseline selection follows directly from our primarily targets under limited interventions. [2] notes that deep models (foundation models) underperform linear baselines evaluate attention-based architecture. Linear or GLM models do not provide (i) a disentangled latent space separating invariant and variant factors, or (ii) an interpretable latent DAG, which are the core objects of interest in our CRL setting.
>
> ---
>
> `W&Q 2`
>
> •***Why no GRN?***
>
> The DAG learned by our model is not a gene-level GRN; it is a causal structure defined over latent programs in $\mathbf{z}_\nu$, rather than observed genes. Therefore, GRN inference benchmarks—which evaluate gene-to-gene regulatory edges—are not applicable to this** latent-space representation**.
>
> • ***Clarification on DAG validation***
>
> The DAG in Figure 4a is **not a gene-level GRN** but a **causal graph over the latent programs** $\mathbf{z}_\nu$. Following the validation strategy used in [4], we assess the latent DAG through its biological fidelity: each latent program is mapped to its associated genes (Appendix C.3), and the resulting causal edges are evaluated against known regulatory mechanisms.
>
> ---
>
> `W&Q 3`
>
> • ***Datasets***
>
> We evaluate our method on **both synthetic and real datasets**. To our knowledge, the only widely used open sourced dataset that supports the **single-gene → unseen double-gene OOD (zero-short)** setting that our method is specifically designed to address in the CRL related work literature[1, 4]. Other datasets, such as Replogle2022, contain only single-gene perturbations and therefore cannot be used for the combinatorial OOD setting studied here. Notably, the unseen single-gene OOD setting mentioned by the reviewer is an even more challenging problem, as it requires generalizing to entirely novel biological mechanisms; we view this as an important direction for future work. In addition, for a** DE-gene–focused** analysis, we added a DE genes dataset.
>
> • ***Latent Causal Structure ≠ Gene-Level Prediction***
>
> Our goal is fundamentally a **latent-space causal representation** task, not a gene-space prediction task. The model is designed to learn a **disentangled causal latent structure**. Accordingly, we evaluate (i) **population-level transcriptome fidelity** using $R^2$, and (ii) the **structural coherence** of the learned latent causal graph.
>
> ---
>
> `W&Q 4`
>
> As detailed in Footnote 6 and Appendix B.4, we compute $R^2$ at the population-average level: it is simply the **square of the Pearson correlation coefficient (square of Pearson Delta [2])**.
>
> In addition, for a DE-gene–focused analysis, we preprocess Norman2019 into a **DE genes dataset**, and we report the results in Appendix C.6.
>
> ---
>
> `W&Q 5`
>
> •***SAMS-VAE***
>
> We agree that SAMS-VAE[3] is an important representative of causal-inference-based perturbation modeling. We have now included a complete experimental comparison against SAMS-VAE, and our method consistently achieves superior performance on real perturbation benchmarks.
>
> •***Clarification on Theoretical Assumptions and Identifiability***
>
> We acknowledge that our assumptions may look idealized, but they are **mathematical necessities for non-linear identifiability in CRL**. The central goal of identifiable CRL is to make the recovery of a **sparse, structurally meaningful latent causal graph** tractable. Our key theoretical contribution (Thm.~3.1) is to provide identifiability guarantees in a **limited-perturbation regime**, where many latent factors remain unperturbed—a strictly milder and more realistic condition than prior work that assumes all latent dimensions are perturbed. In future work, we aim to further relax these assumptions to better align with biological complexity.
>
>
> [1] de la Fuente J, Lehmann R, Ruiz-Arenas C, et al. Interpretable Causal Representation Learning for Biological Data in the Pathway Space[J]. arXiv preprint arXiv:2506.12439, 2025.
>
> [2] Ahlmann-Eltze C, Huber W, Anders S. Deep-learning-based gene perturbation effect prediction does not yet outperform simple linear baselines[J]. Nature Methods, 2025: 1-5.
>
> [3] Bereket M, Karaletsos T. Modelling cellular perturbations with the sparse additive mechanism shift variational autoencoder[J]. Advances in Neural Information Processing Systems, 2023, 36: 1-12
>
> [4] Zhang J, Greenewald K, Squires C, et al. Identifiability guarantees for causal disentanglement from soft interventions[J]. Advances in Neural Information Processing Systems, 2023, 36: 50254-50292.

---

### Official Review · Reviewer_7EUD · 2025-10-25

**Soundness:** 3
**Presentation:** 3
**Contribution:** 2
**Rating:** 2
**Confidence:** 3

**Summary:**

This paper proposes a latent variable model that disentangles perturbation-induced effects from invariant background cellular programs. It partitions the latent space into a "variant" subspace for perturbation effects and an "invariant" subspace for background programs. The authors provide theoretical identifiability guarantees and implement this idea in a model called Contrastive DAG Variational Autoencoder (CDAG-VAE). Experiments show the model improves prediction for unseen combinatorial perturbations.

**Strengths:**

- The method is supported by a theoretical foundation with identifiability guarantees, which adds rigor to the approach.
- The model demonstrates the ability to recover some underlying causal structures.

**Weaknesses:**

- Weak baselines. It is unclear how the results compare to well-established benchmarks in [1-2]. High R2 for linear baselines has already been noted in [2]. It is unclear how it improves over non-VAE approaches.
- The graph shown in Fig. 4a is not the “real” result since it cherry-picks genes from the large gene set corresponding to latent variables. What happens to other genes in the program? Does the co-existence indicate correlation, causality, or are there intermediate genes within these arrows? What about other genes uncovered? In summary, I doubt if the current plot represents biological meaningful regulatory networks, and this should be clarified.
- The identified graph can be vulnerable, and it remains unclear whether it is robust to different random seeds. Overall, the empirical evidence shown is not sufficient.
- The plot of the unperturbed space appears problematic. They should be drawn in an overlapped manner. The current plot in Fig. 4(b) does not demonstrate whether the four perturbations indeed overlap.
- A quick search reveals a relevant paper in 2024 [3] that demonstrates a very similar idea of disentangling unperturbed space and perturbation space with identifiability guarantee. A discussion should be provided on the connection between the works.

[1] Adduri, Abhinav K., et al. "Predicting cellular responses to perturbation across diverse contexts with State." bioRxiv(2025): 2025-06.
[2] Ahlmann-Eltze, Constantin, Wolfgang Huber, and Simon Anders. "Deep-learning-based gene perturbation effect prediction does not yet outperform simple linear baselines." Nature Methods (2025): 1-5.
[3] Dong, Mingze, et al. "Scaling deep identifiable models enables zero-shot characterization of single-cell biological states." bioRxiv (2024): 2023-11.

**Questions:**

See weaknesses.

---

> ### Author Response · Authors · 2025-11-19
>
> `W&Q 1`
>
> • **Different objectives.**
>
> We agree that [1] and [2] are important benchmarks for *predictive* modeling of perturbation responses. However, our work addresses a different primary objective. Methods such as State [1] and the models benchmarked in [2] are powerful **black-box** predictors (foundation models). In contrast, our goal is **identifiable causal representation learning (CRL)**: we aim to disentangle perturbation-responsive factors ($\mathbf{z}_ \nu$) from invariant background programs ($\mathbf{z}_\iota$) and to recover a **latent causal graph** over these programs. These structural and identifiability properties are not targeted, nor evaluated, in [1–2].
>
> • **On linear baselines and non-VAE approaches.**
>
> Linear or GLM models do not provide (i) a disentangled latent space separating invariant and variant factors, or (ii) an interpretable latent DAG, which are the core objects of interest in our CRL setting. Thus, while they are informative for forecasting expression changes, they are not well-suited as **primary baselines for causal structure recovery**.
>
> • **Justification of VAE-based baselines and inclusion of SAMS-VAE.**
>
> Because our contribution lies in **identifiability guarantees and latent causal structure**, we primarily compare against methods from the **identifiable VAE / CR** work, by leveraging perturbation labels as auxiliary variables to explicitly learn a structured causal generative model. In response to the reviewer’s concern, we have additionally included **SAMS-VAE** [3] as a strong causal-inference–based baseline.
>
> ---
>
> `W&Q 2`
>
> We agree that interpreting latent causal structure is challenging, and we clarify that **Figure 4a** is intended as a representative, consensus view, supported by additional analyses in Appendix C.3.
>
> • **Robustness across random seeds.**
>
> We explicitly evaluated robustness by running the model with multiple random seeds. We observed that the **causal structure between latent programs** (Figure 4a) is highly consistent across seeds, and key regulatory backbones reliably reappear. Figure 4a thus summarizes edges that are recurrently discovered across runs, rather than the outcome of a single seed. At the level of program–gene associations (Table 6), some genes do move between closely related programs when the seed changes, which is expected given the difficulty of recovering a latent causal graph from finite, noisy data. Our goal, in line with most CRL work, is to recover the **dominant functional modules and their causal relations**, and these are robust across seeds in our experiments.
>
>
> • **Biological plausibility.**
>
> Finally, we **cross-checked** the most prominent edges in Fig. 4a against the literature. As summarized in Table 7, the model recovers several well-established regulatory relationships.
>
> ---
>
> `W&Q 3`
> See W&Q 2.
>
> ---
>
> `W&Q 4`
> In the main text, Figure 4(b) was deliberately plotted as separate panels while keeping the same invariant latent space $\mathcal{Z}_ {\iota}$ fixed, in order to illustrate how each perturbation is distributed in that shared space when viewed individually. We provide an overlapped visualization in **Appendix C.4, Figure 9**, where we plot all perturbation conditions together in the same t-SNE embedding of $\mathcal{Z}_ {\iota}$.
>
> ---
>
> `W&Q 5`
> We agree **scShift proposed by Dong et al** [3] shares our high-level goal of disentangling an invariant “background” subspace from perturbation-specific effects under identifiability assumptions.
>
> Our approach, however, differs from scShift in three key aspects: **(i) Causal Structure**: cDAG-VAE includes an explicit latent SCM and learns a DAG end-to-end (vs. scShift’s post-hoc causal analysis on embeddings), **(ii) Identifiability mechanism**: our identifiability guarantees (Thm. 3.1) are based on contrastive alignment between perturbed and matched unperturbed cells (vs. scShift’s sparse mechanism-shift / probabilistic $L_0$ assumption), and **(iii) Task setting**: cDAG-VAE targets predictive modeling of perturbation outcomes in a limited-intervention OOD setting (vs. scShift’s atlas-level state integration). We now explicitly discuss and cite scShift in the revised Related Work section.
>
>
>
> [1] Adduri, Abhinav K., et al. "Predicting cellular responses to perturbation across diverse contexts with State." bioRxiv(2025): 2025-06.
>
> [2] Ahlmann-Eltze, Constantin, Wolfgang Huber, and Simon Anders. "Deep-learning-based gene perturbation effect prediction does not yet outperform simple linear baselines." Nature Methods (2025): 1-5.
>
> [3] Bereket M, Karaletsos T. Modelling cellular perturbations with the sparse additive mechanism shift variational autoencoder[J]. Advances in Neural Information Processing Systems, 2023, 36: 1-12
>
> [4] Dong M, Agrawal K, Fan R, et al. Scaling deep identifiable models enables zero-shot characterization of single-cell biological states[J]. bioRxiv, 2024: 2023.11. 11.566161.

---

> > ### Comment · Reviewer_7EUD · 2025-11-19
> >
> > Thanks for the rebuttal that addresses some of my points. But I think the response to the two most important points are not satisfactory (as of yet):
> > 1. While the CRL approach indeed offers causal structures, all current quantitative benchmarking (especially for real data) can be done for linear or foundation models. If your model indeed works, they would substantially outperform these black-box models since the CRL in principle fully addresses the generative process.
> > 2. There are no quantitative results on the inferred causal structure. I am pretty sure that you can generate a (more?) reasonable causal graph using ChatGPT, but that would be almost meaningless. Therefore the current interpretation is really not convincing.
> > 3. It is still highly unclear to me that: Does the co-existence indicate correlation, causality, or are there intermediate genes within these arrows? What about other genes uncovered? The practical utility of the method appears very limited to me if this is not conceptually or empirically validated with strong evidence.

---

> ### Author Response · Authors · 2025-12-01
>
> Thanks for the reviewer's reply, thanks a lot for the reviewer's feedback on our submission.
>
> `Q1`
>
> We have added extra experiments in **Appendix C.7** comparing our method against **(i)**  **additive linear model** (the standard linear baseline for Norman2019 task in [2]) and **(ii)** **GEARS** [5].
>
> All methods are evaluated in the same **single-gene → unseen double-gene OOD** setting that our work targets. In this regime, our model **consistently outperforms GEARS across key metrics** on double-gene perturbations, indicating that the proposed CRL architecture is competitive with GEARS. Compared to the additive linear baseline, we find that linear models remain surprisingly strong on some global metrics—consistent with recent findings—but our method **matches** (RMSE) or **surpasses** ($R^2$) the additive model on the key OOD evaluation measures, particularly in regimes where non-additive genetic interactions become important. For the detailed discussion regarding the additive model and GEARS, please refer to **C.7** of our updated supplementary materials.
>
> ---
>
> `Q2`
>
> First, we have to clarify again that our work is explicitly framed in the setting of **CAUSAL REPRESENTATION LEARNING**. The central object of study is the DGP in a latent space: we assume the existence of latent causal variables governed by a SCM, and we learn a generative mapping from this latent SCM to the observed gene expression. Assumptions in the latent space are therefore a premise of CRL itself, not an arbitrary modeling choice specific to our method [7].
>
> For real large-scale biological systems, a **ground-truth** gene-level causal graph is not available; this is precisely why the CRL and causal discovery literature use **simulated data** to quantitatively assess structural recovery. In line with this, we first evaluate our model on **synthetic data** where the underlying SCM is known. As reported in Table 1, the model accurately recovers both the latent factors and their directed edges (high $R^2$ for $\mathbf{z}_\nu$ and low error on the learned DAG), demonstrating that the proposed architecture can indeed identify the intended latent causal structure when ground truth is available.
>
> Our goal is not to infer a fully resolved gene-level GRN, but a **coarse-grained causal structure over latent cellular programs**. Following [6], we construct program nodes by **hard assignment via maximal intervention effect**: each gene is assigned to the latent dimension whose activation best explains its perturbation response. Biologically, this groups genes with similar downstream effects into the same functional module (e.g., cell-cycle, EMT), and the edges in Figure 4a are the learned directed dependencies between these modules in the latent SCM.
>
> This addresses the reviewer’s specific questions:
>
> The edges are parameters of a **structural causal model in latent space** (Eq.~(2)); an arrow $z_i \to z_j$ encodes a directed functional dependence in the generative process, not just co-existence.
> - **Intermediate genes within arrows.** Each node aggregates many genes, so an edge $z_i \to z_j$ is a **coarse-grained causal relationship between programs**. It may correspond to multi-step gene-level regulation; this abstraction is deliberate, to capture dominant information flow without pretending to resolve every intermediary molecule.
> - **Genes not strongly assigned to a program.** Genes not strongly associated with any program are absorbed into the invariant/background representation $\mathbf{z}_\iota$ or treated as residual noise, so that the latent graph focuses on **driver mechanisms** rather than exhaustively annotating all 5,000 genes.
>
> In summary, the latent causal structure we learn is (i) **quantitatively validated** on simulations with known ground truth in the latent SCM, (ii) **supported on real data** by improved OOD perturbation prediction over linear and deep baselines, and (iii) **biologically interpretable** as a coarse-grained program-level SCM, which is the appropriate level of causal abstraction in the CRL setting we study.
>
>
>
>
> [5] Roohani Y, Huang K, Leskovec J. Predicting transcriptional outcomes of novel multigene perturbations with GEARS[J]. Nature Biotechnology, 2024, 42(6): 927-935.
>
> [6] Zhang J, Greenewald K, Squires C, et al. Identifiability guarantees for causal disentanglement from soft interventions[J]. Advances in Neural Information Processing Systems, 2023, 36: 50254-50292.
>
> [7] Schölkopf B, Locatello F, Bauer S, et al. Toward causal representation learning[J]. Proceedings of the IEEE, 2021, 109(5): 612-634.

---

### Official Review · Reviewer_7SjM · 2025-11-01

**Soundness:** 3
**Presentation:** 2
**Contribution:** 2
**Rating:** 4
**Confidence:** 3

**Summary:**

This paper introduces a generative model designed to predict single-cell gene expression responses to unseen combinatorial perturbations. The novelty is an explicit partitioning of the latent space into an invariant subspace (which is unaffected by perturbations) and a variant subspace which captures the perturbation-induced effects. The authors provide theoretical guarantees that this model can, under a set of assumptions, achieve identifiability and thus uniquely recover the true underlying invariant programs and causal factors.

**Strengths:**

The model demonstrates sota performance in predicting the effects of unseen double-gene perturbations.

**Weaknesses:**

The model's identifiability guarantees rely on several strong, simplifying assumptions that are unlikely to hold in real systems. This creates a significant gap between the theoretical claims (and the consequent model design and guarantees) and its practical reliability.

1. The model assumes a linear SCM with Gaussian noise in the latent space, which is an oversimplification as biological networks are highly non-linear, and single-cell data is count-based (not Gaussian). This risks that all non-linear causal effects are modeled incorrectly, impacting the disentanglement the model claims to achieve.

2. The identifiability proof requires the decoder to be a diffeomorphism, but the mapping from a low dimensional latent space to a high dimension expression profile is almost certainly not invertible.

3. The theory itself is data-dependent, requiring perturbation richness (Assumption 3.3). This means the training data must contain single-gene perturbations diverse enough to excite all causal pathways, which is not usually met.

In other words, the theoretical claims are built on a simplified, linear-Gaussian abstraction, and therefore, the validity of its guarantees in the face of non-linear, non-Gaussian, and real biological data remains an open question.

**Questions:**

See weaknesses. Especially, can you test how much the data obey to your assumption?

How accurate is in predicting differentially expressed genes? Can you also provide a detailed breakdown of which specific perturbations are predicted better or worse than others and do you have an intuition on why?

---

> ### Author Response · Authors · 2025-11-19
>
> We agree that our identifiability analysis relies on simplifying latent-space assumptions. These assumptions are **not biological claims but standard mathematical requirements** in causal representation learning (CRL) to make the recovery of a sparse latent causal graph identifiable. Importantly, linearity and Gaussian noise are assumed only in the **latent SCM**, while the mapping to gene expression is modeled through a **fully nonlinear decoder**, allowing us to capture the rich nonlinear structure of real single-cell data. Our theory also relaxes prior CRL results by providing guarantees in the more realistic limited-perturbation regime, rather than assuming all latent pathways are perturbed. While the assumptions may seem idealized, they enable mathematical tractability and rigorous identifiability guarantees that clarify what structure the learning process can theoretically recover. Empirically, the architecture inspired by this theory, especially the contrastive alignment term—remains robust on real, nonlinear, and non-Gaussian data (**Table 1, Table 2**). We acknowledge the remaining gap between theory and biological complexity and will continue to relax these assumptions in future work.
>
> ---
> `W&Q 1: How much the data obey to your assumption?`
>
> We cannot directly test whether the **unobserved latent space** strictly follows Gaussianity or a linear SCM. Instead, we evaluate the **central testable consequence** of our theory: Theorem 3.1 predicts that the contrastive alignment term is crucial for separating the invariant subspace $\mathcal{Z}_ {\iota}$ from the variant subspace $\mathcal{Z}_ {\nu}$. This is consistently supported by (i) simulations with ground-truth latents, where removing contrastive alignment substantially degrades recovery of $\mathcal{Z}_ {\iota}$, (ii) visualizations on real data (Fig. 4b, App. C.4), where cells from all perturbations remain well mixed in $\mathcal{Z}_ {\iota}$, and (iii) ablation studies, where turning off contrastive alignment causes $\mathcal{Z}_ {\iota}$ to collapse and performance to drop. Taken together, these results indicate that, even if the latent assumptions only hold approximately, the key implication of the theory—successful disentanglement of invariant and perturbation-responsive factors—is indeed realized in practice.
>
> ---
>
> `W&Q 2: How accurate is in predicting differentially expressed genes? `
>
> Our current evaluation focuses on **global transcriptome fidelity** rather than explicit DE-gene classification. On real data, we compute population-average $R^2$ (Pearson-$R^2$) between the predicted and observed **mean expression vectors** over ~5,000 genes, which measures how well we recover the full expression profile, not just a small subset. We agree that an explicit DE gene evaluation would be a valuable complementary analysis, in addition, for a DE-gene–focused analysis, we preprocess Norman2019 into a **20-dimensional differentially expressed–gene dataset**, and we report the corresponding results in Appendix C.6.
>
> ---
> `W&Q 3: Can you also provide a detailed breakdown of which specific perturbations are predicted better or worse than others and do you have an intuition on why?`
>
> As clarified in our response to W\&Q 2, we now include DE-gene–based experiments and a perturbation-wise analysis in Appendix C.6. At a high level, we observe that **single-gene perturbations and many double-gene combinations are predicted very accurately**: the $R^2$ on the top 20 most differentially expressed genes closely matches the global $R^2$, and the RMSE on DE genes is typically lower than on all $\sim 5{,}000$ genes. A subset of double-gene perturbations shows relatively lower $R^2$ on the DE-gene subspace, while the global $R^2$ remains high.
>
> Our intuition is that this reflects the different roles of the invariant and variant latent subspaces. The **invariant subspace**  $\mathbf{z}_ {\iota}$ stably captures background programs and supports robust recovery of the global transcriptomic state across all perturbations, including unseen combinations. In contrast, the **variant subspace** $\mathbf{z}_ {\nu}$ must perform **zero-shot extrapolation** for unseen double-gene pairs. Perturbations whose combined effect is close to a compositional combination of their single-gene responses are predicted best, whereas pairs that induce **strong, non-additive (synergistic or antagonistic) interactions** are naturally harder, as such emergent effects are not directly observed in the single-gene training data.

---

### Meta-Review · Area_Chair_2MKD · 2026-01-05

**Summary:**

This submission presents a disentangled generative model, cDAG-VAE, to predict single-cell gene perturbation responses. The authors also provided the identifiability analysis under the latent structural causal model (SCM) assumptions for the proposed causal representation learning (CRL). Experiments on combinatorial perturbations (single- and double-gene perturbations from Perturbseq or Normal2019 benchmark) were presented to demonstrate the efficacy of cDAG-VAE.

**Reviewer Concerns:**

Based on the available reviewer-author discussions, the authors may have to address the following concerns:

1. The identifiability theoretical guarantee with strong latent SCM assumptions might not be valid in cDAG-VAE with underdetermined decoder. Reviewer 7SjM asked for justification as well as intuitive explanation how latent linear SCM identifiability guarantee leads to the claimed performance gains. The authors used empirical results but did not clearly explain or provide convincing examples where CRL helps unseen combinatorial perturbation prediction.

2. Almost all reviewers raised the concerns on limited experimental validation, including the potential bias of presented perturbation prediction task settings as well as lack of benchmarking with more up-to-date state-of-the-arts. Besides including more recent methods, the authors may want to provide more ablation studies to clearly demonstrate, for example, how the corresponding components in cDAG-VAE, including CRL with DAG, contrastive training, etc., contribute to the prediction performances. The 'cherry-picking' and biological plausibility concerns on by reviewer 7EUD may need to be carefully addressed.

**Reviewer Scores:**

Based on the available discussions, the reviewers may keep their current scores.

---

### Decision · Program_Chairs · 2026-01-26

Reject